# Robust Stochastic Optimization via Gradient Quantile Clipping

**Ibrahim Merad**                                                                    *imerad@lpsm.paris*
*LPSM, UMR 8001*
*Université Paris Cité, Paris, France*

**Stéphane Gaïffas**                                                          *stephane.gaiffas@lpsm.paris*
*LPSM, UMR 8001*
*Université Paris Cité, Paris, France*
*DMA, École normale supérieure*

**Reviewed on OpenReview:** *https://openreview.net/forum?id=HCRkV3kxHW*

## Abstract

We introduce a clipping strategy for Stochastic Gradient Descent (SGD) which uses quantiles of the gradient norm as clipping thresholds. We prove that this new strategy provides a robust and efficient optimization algorithm for smooth objectives (convex or non-convex), that tolerates heavy-tailed samples (including infinite variance) and a fraction of outliers in the data stream akin to Huber contamination. Our mathematical analysis leverages the connection between constant step size SGD and Markov chains and handles the bias introduced by clipping in an original way. For strongly convex objectives, we prove that the iteration converges to a concentrated distribution and derive high probability bounds on the final estimation error. In the non-convex case, we prove that the limit distribution is localized on a neighborhood with low gradient. We propose an implementation of this algorithm using rolling quantiles which leads to a highly efficient optimization procedure with strong robustness properties, as confirmed by our numerical experiments.

## 1 Introduction

Stochastic gradient descent (SGD) (Robbins & Monro, 1951) is the core optimization algorithm at the origin of most stochastic optimization procedures (Kingma & Ba, 2014; Defazio et al., 2014; Johnson & Zhang, 2013). SGD and its variants are ubiquitously employed in machine learning in order to train most models (Kushner & Yin, 2003; Benveniste et al., 2012; Lan, 2020; Shalev-Shwartz et al., 2007; Bottou et al., 2018; Ma et al., 2018). The convergence properties of SGD are therefore subjects of major interest.

Early studies of SGD convergence generally relied on strong assumptions such as bounded domain (Shalev-Shwartz et al., 2009) or uniformly bounded gradient variance (Rakhlin et al., 2011) and obtained error bounds in expectation. With the recent resurgence of interest for robust statistics (Hsu & Sabato, 2016; Diakonikolas et al., 2019; Lecué & Lerasle, 2017; Prasad et al., 2018), variants of SGD based on clipping are shown to be robust to heavy-tailed gradients (Gorbunov et al., 2020; Tsai et al., 2022), where the gradient samples are only required to have a finite variance. The latter requirement has been further weakened to the existence of a $q$-th moment for some $q > 1$ in (Sadiev et al., 2023; Nguyen et al., 2023). In this paper, we go further and show that another variant of clipped SGD with proper thresholds is robust both to heavy tails *and* outliers in the data stream.

Robust statistics appeared in the 60s with the pioneering works of Huber, Tukey and others (Tukey, 1960; Huber, 1992; 1972; Rousseeuw & Hubert, 2011; Hampel, 1971). More recently, the field found new momentum thanks to a series of works about robust scalar mean estimation (Catoni, 2012; Alon et al., 1996; Jerrum

et al., 1986; Lugosi & Mendelson, 2021) and the more challenging multidimensional case (Hopkins, 2020; Catoni & Giulini, 2018; Lugosi & Mendelson, 2019; Minsker, 2015; Cherapanamjeri et al., 2019; Depersin & Lecué, 2022; Lei et al., 2020; Diakonikolas et al., 2020). These paved the way to the elaboration of a host of robust learning algorithms (Holland & Ikeda, 2019; Prasad et al., 2018; Lecué & Lerasle, 2017; Liu et al., 2020; Pensia et al., 2020) which have to date overwhelmingly focused on the batch learning setting. We consider the setting of streaming stochastic optimization (Bottou & Cun, 2003; Bottou & Lecun, 2005; McMahan et al., 2013), which raises an additional difficulty coming from the fact that algorithms can see each sample only once and must operate under an $\mathcal{O}(d)$ memory and complexity constraint for $d$-dimensional optimization problems. A limited number of papers (Tsai et al., 2022; Nazin et al., 2019; Diakonikolas et al., 2022) propose theoretical guarantees for robust algorithms learning from streaming data.

This work introduces such an algorithm that learns from data on the fly and is robust both to heavy tails and outliers, with minimal computational overhead and sound theoretical guarantees.

We consider the problem of minimizing a smooth objective

$$\min_{\theta \in \mathbb{R}^d} \mathcal{L}(\theta) := \mathbb{E}_\zeta[\ell(\theta, \zeta)] \tag{1}$$

using observations $G(\theta, \zeta_t)$ of the unknown gradient $\nabla \mathcal{L}(\theta)$, based on samples $(\zeta_t)_{t \geq 0}$ received sequentially that include corruptions with probability $\eta < 1/2$. Formulation (1) is common to numerous machine learning problems where $\ell$ is a loss function evaluating the fit of a model with parameters $\theta$ on a sample $\zeta$, the expectation $\mathbb{E}$ is w.r.t the unknown uncorrupted sample distribution.

We introduce quantile-clipped SGD (QC-SGD) which uses the iteration

$$\theta_{t+1} = \theta_t - \alpha_{\theta_t} \beta G(\theta_t, \zeta_t) \quad \text{with} \quad \alpha_{\theta_t} = \min\left(1, \frac{\tau_{\theta_t}}{\|G(\theta_t, \zeta_t)\|}\right), \tag{2}$$

where $\beta > 0$ is a constant step size and $\alpha_{\theta_t}$ is the clipping factor with threshold chosen as the $p$-th quantile $\tau_{\theta_t} = Q_p(\|\widetilde{G}(\theta_t, \zeta_t)\|)$ with $\widetilde{G}(\theta_t, \zeta_t)$ an uncorrupted sample of $\nabla \mathcal{L}(\theta_t)$ and $p \in (0, 1)$ (details will follow). Quantiles are a natural choice of clipping threshold which allows to handle heavy tails (Rothenberg et al., 1964; Bloch, 1966) and corrupted data. For instance, the trimmed mean offers a robust and computationally efficient estimator of a scalar expectation (Lugosi & Mendelson, 2021). Since the quantile $Q_p(\|\widetilde{G}(\theta_t, \zeta_t)\|)$ is non-observable, we introduce a method based on rolling quantiles in Section 5 which keeps the procedure $\mathcal{O}(d)$ both memory and complexity-wise. The main benefit of QC-SGD 2 is to grant robustness to the presence of a proportion $\eta < 1/2$ of corruptions in the stream of gradient samples. This could not be achieved by previous clipped SGD methods (Gorbunov et al., 2020; Tsai et al., 2022; Sadiev et al., 2023; Nguyen et al., 2023). We also show that iteration (2) is adaptive to heavy-tailed gradient variance and converges to a limit distribution with strong concentration properties.

**Contributions.** Our main contributions are as follows:

- For small enough $\eta$ and well-chosen $p$, we show that, whenever the optimization objective is smooth and strongly convex, QC-SGD converges *geometrically* to a limit distribution such that the deviation around the optimum achieves the *optimal* dependence on $\eta$.

- In the non-corrupted case $\eta = 0$ and with a strongly convex objective, we prove that a coordinated choice of $\beta$ and $p$ ensures that the limit distribution is sub-Gaussian with constant of order $\mathcal{O}(\sqrt{\beta})$. In the corrupted case $\eta > 0$, the limit distribution is sub-exponential.

- For a smooth objective (non-convex) whose gradient satisfies an identifiability condition, we prove that the total variation distance between QC-SGD iterates and its limit distribution vanishes sub-linearly. In this case, the limit distribution is such that the deviation of the objective gradient is optimally controlled in terms of $\eta$.

- Finally, we provide experiments to demonstrate that QC-SGD can be easily and efficiently implemented by estimating $Q_p(\|\widetilde{G}(\theta_t, \zeta_t)\|)$ with rolling quantiles. In particular, we show that the iteration is indeed robust to heavy tails and corruption on multiple stochastic optimization tasks.

Our theoretical results are derived thanks to a modelling through Markov chains and hold under an $L_q$ assumption on the gradient distribution with $q > 1$.

**Related works.** Convergence in distribution of the Markov chain generated by constant step size SGD, relatively to the Wasserstein metric, was first established in (Dieuleveut et al., 2020). Another geometric convergence result was derived in (Yu et al., 2021) for non-convex, non-smooth, but quadratically growing objectives, where a convergence statement relatively to a weighted total variation distance is given and a CLT is established. These papers do not consider robustness to heavy tails or outliers. Early works proposed stochastic optimization and parameter estimation algorithms which are robust to a wide class of noise of distributions (Martin & Masreliez, 1975; Polyak & Tsypkin, 1979; 1981; Price & VandeLinde, 1979; Stanković & Kovačević, 1986; Chen et al., 1987; Chen & Gao, 1989; Nazin et al., 1992), where asymptotic convergence guarantees are stated for large sample sizes. Initial evidence of the robustness of clipped SGD to heavy tails was given by (Zhang et al., 2020) who obtained results in expectation. Subsequent works derived high-confidence sub-Gaussian performance bounds under a finite variance assumption (Gorbunov et al., 2020; Tsai et al., 2022) and later under an $L_q$ assumption (Sadiev et al., 2023; Nguyen et al., 2023) with $q > 1$. A similar SGD clipping scheme to (2) is presented in (Seetharaman et al., 2020), however, in contrast to our work, they do not consider the robust setting and focus on experimental study while we also provide theoretical guarantees.

Robust versions of Stochastic Mirror Descent (SMD) are introduced in (Nazin et al., 2019; Juditsky et al., 2023). For a proper choice of the mirror map, SMD is shown to handle infinite variance gradients without any explicit clipping (Nemirovskij & Yudin, 1983; Vural et al., 2022). Finally, (Diakonikolas et al., 2022) study heavy-tailed and outlier robust streaming estimation algorithms of the expectation and covariance. On this basis, robust algorithms for linear and logistic regression are derived. However, the involved filtering procedure is hard to implement in practice and no numerical evaluation of the considered approach is proposed.

**Agenda.** In Section 2 we set notations, state the assumptions required by our theoretical results and provide some necessary background on continuous state Markov chains. In Section 3, we state our results for strongly convex objectives including geometric ergodicity of QC-SGD (Theorem 1), characterizations of the limit distribution and deviation bounds on the final estimate. In Section 4, we remove the convexity assumption and obtain a weaker ergodicity result (Theorem 2) and characterize the limit distribution in terms of the deviations of the objective gradient. Finally, we present a rolling quantile procedure in Section 5 and demonstrate its performance through a few numerical experiments on synthetic and real data.

## 2 Preliminaries

The model parameter space is $\mathbb{R}^d$ endowed with the Euclidean norm $\|\cdot\|$, $\mathcal{B}(\mathbb{R}^d)$ is the Borel $\sigma$-algebra of $\mathbb{R}^d$ and we denote by $\mathcal{M}_1(\mathbb{R}^d)$ the set of probability measures over $\mathbb{R}^d$. We assume throughout the paper that the objective $\mathcal{L}$ is smooth.

**Assumption 1.** *The objective $\mathcal{L}$ is $L$-Lipschitz-smooth, namely*

$$\mathcal{L}(\theta') \leq \mathcal{L}(\theta) + \langle \nabla \mathcal{L}(\theta), \theta' - \theta \rangle + \frac{L}{2}\|\theta - \theta'\|^2$$

*with $L < +\infty$ for all $\theta, \theta' \in \mathbb{R}^d$.*

The results from Section 3 below use the following

**Assumption 2.** *The objective $\mathcal{L}$ is $\mu$-strongly convex, namely*

$$\mathcal{L}(\theta') \geq \mathcal{L}(\theta) + \langle \nabla \mathcal{L}(\theta), \theta' - \theta \rangle + \frac{\mu}{2}\|\theta - \theta'\|^2$$

*with $\mu > 0$ for all $\theta, \theta' \in \mathbb{R}^d$.*

An immediate consequence of Assumption 2 is the existence of a unique minimizer $\theta^\star = \arg\min_{\theta \in \mathbb{R}^d} \mathcal{L}(\theta)$. The next assumption formalizes our corruption model.

**Assumption 3** ($\eta$-corruption)**.** *The gradients $(G(\theta_t, \zeta_t))_{t \geq 0}$ used in Iteration (2) are sampled as $G(\theta_t, \zeta_t) = U_t \check{G}(\theta_t) + (1 - U_t)\widetilde{G}(\theta_t, \zeta_t)$ where $U_t$ are i.i.d Bernoulli random variables with parameter $\eta < 1/2$, $\check{G}(\theta_t) \sim \mathcal{D}_{\mathcal{O}}(\theta_t)$ with $\mathcal{D}_{\mathcal{O}}(\theta_t)$ an arbitrary distribution and $\widetilde{G}(\theta_t, \zeta_t) \sim \mathcal{D}_{\mathcal{I}}(\theta_t)$ follows the true gradient distribution and is independent from the past given $\theta_t$.*

Assumption 3 is an online analog of the Huber contamination model (Huber, 1965; 1992) where corruptions occur with probability $\eta$ and where the distribution of corrupted samples $\mathcal{D}_{\mathcal{O}}(\theta_t)$ (outliers) is not fixed and may depend on the current iterate $\theta_t$. On the other hand, $\mathcal{D}_{\mathcal{I}}(\theta_t)$ denotes the distribution of inliers. This notation and dichotomy between inliers and outliers follows the example of (Lecué & Lerasle, 2017; Lecué & Lerasle, 2019). Assumption 3 corresponds to *additive* contamination (Diakonikolas & Kane, 2023, Section 1.2.2) where corruptions are only added to the data. A more general TV-contamination model allowing for true samples to be adversely removed is used in (Diakonikolas et al., 2022). Note however, that the latter mainly focuses on mean estimation. Note also that additive contamination remains realistic since it accounts for invalid entries occurring in the data stream even if it doesn't support entries being targeted and censored. The next assumption requires the true gradient distribution to be unbiased and diffuse.

**Assumption 4.** *For all $\theta$, non-corrupted gradient samples $\widetilde{G}(\theta, \zeta) \sim \mathcal{D}_{\mathcal{I}}(\theta)$ are such that*

$$\widetilde{G}(\theta, \zeta) = \nabla \mathcal{L}(\theta) + \varepsilon_\theta, \tag{3}$$

*where $\varepsilon_\theta$ is a centered noise $\mathbb{E}[\varepsilon_\theta | \theta] = 0$ with distribution $\delta \nu_{\theta,1} + (1 - \delta)\nu_{\theta,2}$ where $\delta > 0$ and $\nu_{\theta,1}, \nu_{\theta,2}$ are distributions over $\mathbb{R}^d$ such that $\nu_{\theta,1}$ admits a density $h_\theta$ w.r.t. the Lebesgue measure satisfying*

$$\inf_{\|\omega\| \leq R} h_\theta(\omega) > \varkappa(R) > 0$$

*for all $R > 0$, where $\varkappa(\cdot)$ is independent of $\theta$.*

In addition to the unbiased property, Assumption 4 imposes that the noise distribution be expressible as the combination of two components, one of which must be diffuse with density satisfying a minorization inequality. Note that this is a weak constraint since it is satisfied, for example, as soon as the noise $\varepsilon_\theta$ admits a density w.r.t. Lebesgue's measure which is positive everywhere. This condition is similar to (Yu et al., 2021, Assumption 2.3) since both find their origin in Markov chain minorization conditions (Meyn & Tweedie, 1993, Section 5.2). These ensure that a chain properly explores its state space and are a common way to prove Markov chain convergence (Rosenthal, 1995b;a; Douc et al., 2004; Meyn & Tweedie, 1994; Baxendale, 2005). Our last assumption formalizes the requirement of a finite moment for the gradient error.

**Assumption 5.** *There is $q > 1$ such that for $\widetilde{G}(\theta, \zeta) \sim \mathcal{D}_{\mathcal{I}}(\theta)$, we have*

$$\mathbb{E}\big[\|\varepsilon_\theta\|^q \mid \theta\big]^{1/q} = \mathbb{E}\big[\big\|\widetilde{G}(\theta, \zeta) - \nabla \mathcal{L}(\theta)\big\|^q \mid \theta\big]^{1/q} \leq A_q \|\theta - \theta^\star\| + B_q \tag{4}$$

*for all $\theta \in \mathbb{R}^d$, where $A_q, B_q > 0$. When $\mathcal{L}$ is not strongly convex, we further assume that $A_q = 0$.*

The bound (4) captures the case of arbitrarily high noise magnitude through the dependence on $\|\theta - \theta^\star\|$. This is consistent with convex optimization problems with $L$-Lipschitz-smooth objectives (Assumption 1) where the norm of the gradient $\|\nabla \mathcal{L}(\theta)\|$ is bounded by $L \cdot \|\theta - \theta^\star\|$. Assumption 5 improves upon the conditions used in (Gorbunov et al., 2020; Tsai et al., 2022; Gorbunov et al., 2023; Nguyen et al., 2023) since these either required a uniformly constant upperbound (independent of $\theta$) or only considered the case $q = 2$ (finite variance). For non-strongly convex $\mathcal{L}$, we require $A_q = 0$ since $\theta^\star$ may not exist.

**Definition 1.** *If $X$ is a real random variable, we say that $X$ is $K$-sub-Gaussian for $K > 0$ if*

$$\mathbb{E} \exp(\lambda^2 X^2) \leq e^{\lambda^2 K^2} \quad for \quad |\lambda| \leq 1/K. \tag{5}$$

*We say that $X$ is $K$-sub-exponential for $K > 0$ if*

$$\mathbb{E} \exp(\lambda|X|) \leq \exp(\lambda K) \quad for \; all \quad 0 \leq \lambda \leq 1/K. \tag{6}$$

The convergence results presented in this paper use the following formalism of continuous state Markov chains. Given a step size $\beta > 0$ and a quantile $p \in (0,1)$, we denote by $P_{\beta,p}$ the Markov transition kernel governing the Markov chain $(\theta_t)_{t \geq 0}$ generated by QC-SGD, so that

$$\mathbb{P}(\theta_{t+1} \in A \mid \theta_t) = P_{\beta,p}(\theta_t, A)$$

for $t \geq 0$ and $A \in \mathcal{B}(\mathbb{R}^d)$. The transition kernel $P_{\beta,p}$ acts on probability distributions $\nu \in \mathcal{M}_1(\mathbb{R}^d)$ through the mapping $\nu \to \nu P_{\beta,p}$ which is defined, for all $A \in \mathcal{B}(\mathbb{R}^d)$, by $\nu P_{\beta,p}(A) = \int_A P_{\beta,p}(\theta, A) d\nu(\theta) = \mathbb{P}(\theta_{t+1} \in A \mid \theta_t \sim \nu)$. For $n \geq 1$, we similarly define the multi-step transition kernel $P_{\beta,p}^n$ which is such that $P_{\beta,p}^n(\theta_t, A) = \mathbb{P}(\theta_{t+n} \in A \mid \theta_t)$ and acts on probability distributions $\nu \in \mathcal{M}_1(\mathbb{R}^d)$ through $\nu P_{\beta,p}^n = (\nu P_{\beta,p}) P_{\beta,p}^{n-1}$. Finally, we define the total variation (TV) norm of a signed measure $\nu$ as

$$2\|\nu\|_{\text{TV}} = \sup_{f:|f| \leq 1} \int f(\theta) \nu(d\theta) = \sup_{A \in \mathcal{B}(\mathbb{R}^d)} \nu(A) - \inf_{A \in \mathcal{B}(\mathbb{R}^d)} \nu(A). \tag{7}$$

In particular, we recover the TV *distance* between $\nu_1, \nu_2 \in \mathcal{M}_1(\mathbb{R}^d)$ as

$$d_{\text{TV}}(\nu_1, \nu_2) = \|\nu_1 - \nu_2\|_{\text{TV}} = \sup_{A \in \mathcal{B}(\mathbb{R}^d)} |\nu_1(A) - \nu_2(A)|.$$

The second equality reflects the fact that the TV distance between two probability measures corresponds to the largest absolute difference between the probabilities they assign to the same event. The TV distance is a broadly used metric to quantify the convergence of Markov chains (Levin & Peres, 2017; Baxendale, 2005; Meyn & Tweedie, 1993; Rosenthal, 1995a) besides the Wasserstein distance (Dieuleveut et al., 2020).

In the next section, we will prove that the Markov chain defined by iteration (2) converges to unique invariant distribution in TV distance. This convergence mode will allow us to extrapolate the properties of the limit distribution on the iterates $\theta_t$ and thus derive non-asymptotic concentration bounds for them, see Corollaries 2 and 1 below.

## 3 Strongly Convex Objectives

We are ready to state our convergence result for the stochastic optimization of a strongly convex objective using QC-SGD with $\eta$-corrupted samples.

**Theorem 1** (Geometric ergodicity). *Let Assumptions 1-5 hold and assume there is a quantile $p \in [\eta, 1-\eta]$ such that*

$$\kappa := (1-\eta)p\mu - \eta L - (1-p)^{-\frac{1}{q}} A_q (1 - p(1-\eta)) > 0. \tag{8}$$

*Then, for a step size $\beta$ satisfying*

$$\beta < \frac{1}{4} \frac{\kappa}{\mu^2 + 24\eta L^2 + 28 A_q^2} \wedge \frac{2}{\mu + L}, \tag{9}$$

*the Markov chain $(\theta_t)_{t \geq 0}$ generated by QC-SGD with parameters $\beta$ and $p$ converges geometrically to a unique invariant measure $\pi_{\beta,p}$: for any initial $\theta_0 \in \mathbb{R}^d$, there is $\rho < 1$ and $M < \infty$ such that after $T$ iterations*

$$\left\| \delta_{\theta_0} P_{\beta,p}^T - \pi_{\beta,p} \right\|_{\text{TV}} \leq M \rho^T \left( 1 + \|\theta_0 - \theta^\star\|^2 \right),$$

*where $\delta_{\theta_0}$ is the Dirac measure located at $\theta_0$.*

The proof of Theorem 1 is given in Appendix D.3 and relies on the geometric ergodicity result of (Meyn & Tweedie, 1993, Chapter 15) for Markov chains with a geometric drift property. A similar result for quadratically growing objectives was established by (Yu et al., 2021) and convergence w.r.t. Wasserstein's metric was shown in (Dieuleveut et al., 2020) assuming gradient co-coercivity. However, robustness was not considered in these works. Theorem 1 establishes the iteration's convergence to a unique invariant measure $\pi_{\beta,p}$. The properties of this limit distribution will be explored in the sequel. The restriction $p \in [\eta, 1-\eta]$ comes from the consideration that other quantiles are not estimable in the event of $\eta$-corruption. Condition (8) is

best interpreted for the choice $p = 1 - \eta$ in which case it translates into $\eta^{1-1/q} \leq \mathcal{O}(\mu/(L + A_q))$ implying that it is verified for $\eta$ small enough within a limit fixed by the problem conditioning. A similar condition with $q = 2$ appears in (Diakonikolas et al., 2022, Theorem E.9) which uses a finite variance assumption.

When (8) is satisfied, one clearly has that $\kappa = \mathcal{O}(\mu)$. Considering $q = 2$ for simplicity and taking the maximum allowed corruption rate in this case $\eta = \mathcal{O}(\mu^2/(L + A_q)^2)$ leads to an upperbound on the step-size $\beta$ of order $\mathcal{O}(\mu/(\mu^2 + A_q^2) \wedge 1/L)$. While the condition $\beta = \mathcal{O}(1/L)$ is standard in smooth optimization, the additional condition in terms of $A_q$ ensures that the noise introduced to the iteration by the gradient samples does not cause it to diverge.

The constants $M$ and $\rho$ controlling the geometric convergence speed in Theorem 1 depend on the parameters $\beta, p$ and the initial $\theta_0$. Among choices fulfilling the convergence conditions, it is straightforward that greater step size $\beta$ and $\theta_0$ closer to $\theta^\star$ lead to faster convergence. However, the dependence in $p$ is more intricate and should be evaluated through the resulting value of $\kappa$. We provide a more detailed discussion about the value of $\rho$ in Appendix C.

The choice $p = 1 - \eta$ appears to be ideal since it leads to optimal deviation of the invariant distribution around the optimum $\theta^\star$ which is the essence of our next statement.

**Proposition 1.** *Assume the same as in Theorem 1 and condition (8) with the choice $p = 1 - \eta$. For step size $\beta$ satisfying (9), $q \geq 2$, and additionally:*

$$\beta \leq \eta^{2-2/q}/\kappa, \tag{10}$$

*for $\theta \sim \pi_{\beta, 1-\eta}$, we have the following upper bound:*

$$\mathbb{E}\|\theta - \theta^\star\|^2 \leq \left(\frac{6\eta^{1-1/q}B_q}{\kappa}\right)^2.$$

Proposition 1 is proven in Appendix D.4. An analogous result holds for $q \in (1, 2)$ but requires a different proof and can be found in Appendix D.5. Proposition 1 may be compared to (Yu et al., 2021, Theorem 3.1) which shows that the asymptotic estimation error can be reduced arbitrarily using a small step size. However, this is impossible in our case since we consider corrupted gradients. The performance of Proposition 1 is best discussed in the specific context of linear regression where gradients are given as $G(\theta, (X, Y)) = X(X^\top \theta - Y)$ for samples $X, Y \in \mathbb{R}^d \times \mathbb{R}$ such that $Y = X^\top \theta^\star + \epsilon$ with $\epsilon$ a centered noise. In this case, a finite moment of order $k$ for the data implies order $k/2$ for the gradient corresponding to an $\eta^{1-2/k}$ rate in Proposition 1. Since Assumption 5 does not include independence of the noise $\epsilon$ from $X$, this corresponds to the negatively correlated moments assumption of (Bakshi & Prasad, 2021) being unsatisfied. Consequently, Proposition 1 is information-theoretically optimal in $\eta$ based on (Bakshi & Prasad, 2021, Corollary 4.2). Nonetheless, the dimension dependence through $B_q$ remains poor since we have $B_q \sim \sqrt{d}$ in general because the Euclidean norm is used in Assumption 5. This dimension dependence may be improvable by using the quantiles $\sup_{\|v\|=1} Q_p\big(|\langle \widetilde{G}(\theta_t, \zeta_t), v\rangle|\big)$ as clipping thresholds and adapting ideas from (Catoni & Giulini, 2018) in the analysis. However, exploring this method is beyond our scope as the involved estimations for all $\|v\| = 1$ would be excessively sample hungry and computationally heavy for stochastic optimization. If the gradient is sub-Gaussian with constant $K$, we would have $B_q \lesssim K\sqrt{q}$ for $q \geq 1$ (see (Vershynin, 2018) for a reference), in which case, the choice $q = \log(1/\eta)$ recovers the optimal rate in $\eta\sqrt{\log(1/\eta)}$ for the Gaussian case.

We now turn to showing strong concentration properties for the invariant distribution $\pi_{\beta,p}$. For this purpose, we restrict the optimization to a bounded and convex set $\Theta \subset \mathbb{R}^d$ and replace Iteration (2) by the projected iteration

$$\theta_{t+1} = \Pi_\Theta\big(\theta_t - \alpha_{\theta_t}\beta G(\theta_t, \zeta_t)\big), \tag{11}$$

where $\Pi_\Theta$ is the projection onto $\Theta$. Assuming that the latter contains the optimum $\theta^\star \in \Theta$, one can check that the previous results continue to hold thanks to the inequality

$$\|\Pi_\Theta(\theta) - \theta^\star\| = \|\Pi_\Theta(\theta) - \Pi_\Theta(\theta^\star)\| \leq \|\theta - \theta^\star\|,$$

which results from the convexity of $\Theta$. The restriction of the optimization to a bounded set allows us to uniformly bound the clipping threshold $\tau_\theta$, which is indispensable for the following result.

**Proposition 2.** *In the setting of Theorem 1, consider projected QC-SGD (11) and let $\overline{\tau} = \sup_{\theta \in \Theta} \tau_\theta$, $D = \mathrm{diam}(\Theta)$ the diameter of $\Theta$ and $\overline{B}_q = A_q D + B_q$.*

- *Consider the non-corrupted case $\eta = 0$ and set the quantile $p$ such that $p \geq 1 - (\beta\mu)^{\frac{q}{2(q-1)}}$. Then, for $\theta \sim \pi_{\beta,p}$, the variable $\|\theta - \theta^\star\|$ is sub-Gaussian in the sense of Definition 1 with constant*

$$K = 4\sqrt{\frac{2\beta(\overline{B}_q^2 + \overline{\tau}^2)}{p\mu}}.$$

- *Consider the corrupted case $\eta > 0$, and set the quantile $p \in [\eta, 1-\eta]$ such that Inequality (8) holds. Then, for $\theta \sim \pi_{\beta,p}$, the variable $\|\theta - \theta^\star\|$ is sub-exponential in the sense of Definition 1 with constant*

$$K = \frac{7\overline{\tau} + (1-p)^{1-1/q}\overline{B}_q}{p\mu}.$$

The proof can be found in Appendix D.6. The strong concentration properties given by Proposition 2 for the invariant distribution appear to be new. Still, the previous result remains asymptotic in nature. High confidence deviation bounds for an iterate $\theta_t$ can be derived by leveraging the convergence in Total Variation distance given by Theorem 1 leading to the following result.

**Corollary 1.** *In the setting of Proposition 2, in the absence of corruption $\eta = 0$, after $T$ iterations, for $\delta > 0$, we have*

$$\mathbb{P}\left(\|\theta_T - \theta^\star\| > 4\sqrt{\overline{B}_q^2 + \overline{\tau}^2}\sqrt{\frac{2\beta\log(e/\delta)}{p\mu}}\right) \leq \delta + \rho^T M\left(1 + \|\theta_0 - \theta^\star\|^2\right).$$

Choosing a smaller step size $\beta$ in Corollary 1 allows to improve the deviation bound. However, this comes at the cost of weaker confidence because of slower convergence due to a greater $\rho$. See Appendix C for a discussion including a possible compromise. Corollary 1 may be compared to the results of (Gorbunov et al., 2020; Tsai et al., 2022; Sadiev et al., 2023; Nguyen et al., 2023) which correspond to $\beta \approx 1/T$ and have a similar dependence on the dimension through the gradient variance. Although their approach is also based on gradient clipping, they use different thresholds and proof methods. In the presence of corruption, the invariant distribution is not sub-Gaussian. This can be seen by considering the following toy Markov chain:

$$X_{t+1} = \begin{cases} \alpha X_t + \xi & \text{w.p.} \quad 1-\eta \\ X_t + \tau & \text{w.p.} \quad \eta \end{cases}$$

where $\alpha < 1, \tau > 0$ are constants and $\xi$ is a positive random noise. Using similar methods to the proof of Theorem 1, one can show that $(X_t)_{t\geq 0}$ converges (for any initial $X_0$) to an invariant distribution whose moments can be shown to grow linearly, indicating a sub-exponential distribution and excluding a sub-Gaussian one. We provide additional details for the underlying argument in Appendix D.7. For the corrupted case, the sub-exponential property stated in Proposition 2 holds with a constant $K$ of order $\overline{\tau}/\mu$, which is not satisfactory and leaves little room for improvement due to the inevitable bias introduced by corruption. Therefore, we propose the following procedure in order to obtain a high confidence estimate, similarly to Corollary 1.

Algorithm 1 uses ideas from (Hsu & Sabato, 2016) (see also (Minsker, 2015; Juditsky et al., 2023)) and combines the collection of *weak* estimators $\left(\theta_T^{(i)}\right)_{i\in[\![N]\!]}$ (only satisfying $L_2$ bounds) into a strong one with sub-exponential deviation. This is done by picking $\theta_T^{(i)}$ which is such that the median of its distances to other estimators $r_{\lfloor N/2 \rfloor}^{(i)}$ is minimal. The aggregated estimator $\widehat{\theta}$ satisfies the high probability bound given in the next result.

**Corollary 2.** *Assume the same as in Theorem 1 and Proposition 1. Consider $\widehat{\theta}$ given by Algorithm 1, with the assumption that the gradient sample sets used for each $\left(\theta_T^{(n)}\right)_{n\in[\![N]\!]}$ in Equation (12) are independent. For $\delta > 0$, if $N \geq 16\log(1/\delta)$ and $T$ satisfies*

$$T \geq N\log(15M(1 + \|\theta_0 - \theta^\star\|^2))/\log(1/\rho),$$

---

**Algorithm 1:** Aggregation of cycling iterates

---

    **Input:** Step size $\beta > 0$, quantile index $p \in (0,1)$, initial parameter $\theta_0 \in \Theta$, horizon $T$ and number of concurrent iterates $N \geq 1$.

**1** Optimize multiple parameters $\theta_t^{(1)}, \ldots, \theta_t^{(N)}$ starting from a common $\theta_0 = \theta_0^{(n)}$ for $n \in [\![N]\!] =: \{1, \ldots, N\}$ and $T$ steps $t = 0, \ldots, T$ using the following cycling iteration:

$$\theta_{t+1}^{(n)} = \begin{cases} \theta_t^{(n)} - \alpha_{\theta_t^{(n)}} \beta G\big(\theta_t^{(n)}, \zeta_t\big) & \text{if } t \equiv n-1 \bmod N, \\ \theta_t^{(n)} & \text{otherwise.} \end{cases} \tag{12}$$

**2** Compute the pairwise distances $r_{i,j} = \big\|\theta_T^{(i)} - \theta_T^{(j)}\big\|$ for $i, j \in [\![N]\!]$.

**3** For $i \in [\![N]\!]$, let $r^{(i)} \in \mathbb{R}_+^N$ be the vector $r_{i,:} := [r_{i,1}, \ldots, r_{i,N}]$ sorted in non decreasing order.

**4** Compute the aggregated estimator as $\widehat{\theta} = \theta_T^{(\widehat{i})}$ with $\widehat{i} = \arg\min_{i \in [\![N]\!]} r_{\lfloor N/2 \rfloor}^{(i)}$.

**5 return** $\widehat{\theta}$

---

*then, with probability at least $1 - \delta$, we have*

$$\big\|\widehat{\theta} - \theta^\star\big\| \leq \frac{27\eta^{1-\frac{1}{q}}\overline{B}_q}{\kappa}. \tag{13}$$

We obtain a high confidence version of the bound in expectation previously stated in Proposition 1. As argued before, the above bound depends optimally on $\eta$. Similar bounds to (13) are obtained for $q = 2$ in (Diakonikolas et al., 2022) for streaming mean estimation, linear and logistic regression. Their results enjoy better dimension dependence but are less general than ours since we handle the case $q \in (1, 2)$ and consider strongly convex objectives more broadly. In addition, our results further extend to non-convex objectives as detailed in the next section. Finally, the implementation of the algorithm in (Diakonikolas et al., 2022) is not straightforward whereas our method is quite easy to use (see Section 5).

## 4 Non-Convex Objectives

In this section, we drop Assumption 2 and consider the optimization of possibly non-convex objectives. Consequently, the existence of a unique optimum $\theta^\star$ and the quadratic growth of the objective are no longer guaranteed. This motivates us to use a uniform version of Assumption 5 with $A_q = 0$ since the gradient is no longer assumed coercive and its deviation moments can be taken as bounded. In this context, we obtain the following weaker (compared to Theorem 1) ergodicity result for QC-SGD.

**Theorem 2** (Ergodicity). *Let Assumptions 1, 3, 4 and 5 hold with $A_q = 0$ (uniformly bounded moments) and let $\mathcal{L}$ be an objective such that $\inf_\theta \mathcal{L}(\theta) > -\infty$ is finite. Let $(\theta_t)_{t \geq 0}$ be the Markov chain generated by QC-SGD with step size $\beta$ and quantile $p \in [\eta, 1-\eta]$. Assume that $p$ and $\beta$ are such that $3p(1-\eta)/4 > L\beta + \eta$ and that the subset of $\mathbb{R}^d$ given by*

$$\left\{ \theta : \frac{1}{2}\big\|\nabla\mathcal{L}(\theta)\big\|^2 \leq \frac{B_q^2\big((1-p)^{-\frac{2}{q}}(L\beta + 2\eta^2) + 2\eta^{2-\frac{2}{q}}\big)}{p(1-\eta)(3p(1-\eta)/4 - L\beta - \eta)} \right\} \tag{14}$$

*is bounded. Then, for any initial $\theta_0 \in \mathbb{R}^d$, there exists $M < +\infty$ such that after $T$ iterations*

$$\big\|\delta_{\theta_0} P_{\beta,p}^T - \pi_{\beta,p}\big\|_{\mathrm{TV}} \leq \frac{M}{T}, \tag{15}$$

*where $\pi_{\beta,p}$ is a unique invariant measure and where $\delta_{\theta_0}$ is the Dirac measure located at $\theta_0$.*

The proof is given in Appendix D.10 and uses ergodicity results from (Meyn & Tweedie, 1993, Chapter 13). Theorem 2 provides convergence conditions for an SGD Markov chain on a smooth objective in a robust setting. We are unaware of anterior results of this kind in the literature. Condition (14) requires that the

set where the true gradient norm is smaller than the estimation error is bounded. This aims to exclude the possibility that the iteration gets trapped within this set and keep using unreliable gradient estimates causing it to diverge. The result is stronger when the upperbound in (14) is smaller. Note that setting $p$ close to $1 - \eta$ increases the clipping threshold and the estimation error as a consequence, making this condition harder to satisfy. On the other hand, using $\beta = \mathcal{O}(1/L)$ and a more conservative value of $p$ makes the upperbound of order $\mathcal{O}(B_q^2)$ and condition 14 easier to satisfy. Observe that, for no corruption ($\eta = 0$), the condition is always fulfilled for some $\beta$ and $p$. Note also that without strong convexity (Assumption 2), convergence occurs at a slower sublinear rate which is consistent with the optimization rate expected for a smooth objective (see (Bubeck, 2015, Theorem 3.3)).

As previously, we complement Theorem 1 with a characterization of the invariant distribution.

**Proposition 3.** *Under the conditions of Theorem 2, assume that the choice $p = 1 - \eta$ is such that the set (14) is bounded. For step size $\beta \le \eta^2/L$, the stationary measure $\theta \sim \pi_{\beta, 1-\eta}$ satisfies*

$$\mathbb{E}\big\|\nabla\mathcal{L}(\theta)\big\|^2 \le \frac{5\eta^{2-\frac{2}{q}}B_q^2}{p(1-\eta)\big(3p(1-\eta)/4 - L\beta - \eta\big)}. \tag{16}$$

The statement of Proposition 3 is clearly less informative than Propositions 1 and 2 since it only pertains to the gradient rather than, for example, the excess risk. This is due to the weaker assumptions that do not allow to relate these quantities. Still, the purpose remains to find a critical point and is achieved up to $\mathcal{O}(\eta^{1-1/q})$ precision according to this result. Due to corruption, the estimation error on the gradient cannot be reduced beyond $\Omega(\eta^{1-1/q})$ (Prasad et al., 2020; Hopkins & Li, 2018; Diakonikolas & Kane, 2019). Therefore, one may draw a parallel with a corrupted mean estimation task, in which case, the previous rate is, in fact, information-theoretically optimal.

## 5 Implementation and Numerical Experiments

The use of the generally unknown quantile $Q_p(\|\widetilde{G}(\theta_t, \zeta_t)\|)$ in QC-SGD constitutes the main obstacle to its implementation. For strongly convex objectives, one may use a proxy such as $a\|\theta_t - \theta_{\mathrm{ref}}\| + b$ with positive $a, b$ and $\theta_{\mathrm{ref}} \in \mathbb{R}^d$ an approximation of $\theta^\star$ serving as reference point. This choice is consistent with Assumptions 1 and 5, see Lemma 2 in Appendix D. For instances of Problem (1) defined with an

---

**Algorithm 2:** Rolling QC-SGD

**Input:** Step size $\beta > 0$, quantile index
$p \in (0, 1)$, initial parameter $\theta_0 \in \mathbb{R}^d$, $\tau_{\mathrm{unif}} > 0$, buffer $B$ of size $S$ and horizon $T$.

**1** Fill $B$ with $S - 1$ values equal to $\tau_{\mathrm{unif}}$.
**2 for** $t = 0 \dots T - 1$ **do**
**3**      Draw a sample $G(\theta_t, \zeta_t)$ and add $\|G(\theta_t, \zeta_t)\|$ to $B$.
**4**      $\widehat{Q}_p \leftarrow \lfloor pS \rfloor$ rank element of $B$.
**5**      $\theta_{t+1} \leftarrow \theta_t - \beta\mathrm{clip}(G(\theta_t, \zeta_t), \widehat{Q}_p)$
**6**      Delete the oldest value in $B$.
**7 return** $\theta_T$

---

asymptotically linear function $\ell$ such as the logistic, hinge or Huber's loss, a constant threshold can be used since the gradient is a priori uniformly bounded, implying the same for the quantiles of its deviations. In practice, we propose a simpler and more direct approach: we use a rolling quantile procedure, described in Algorithm 2. The latter stores the values $(\|G(\theta_{t-j}, \zeta_{t-j})\|)_{1 \le j \le S}$ in a buffer of size $S \in \mathbb{N}^*$ and replaces $Q_p(\|\widetilde{G}(\theta_t, \zeta_t)\|)$ in QC-SGD by an estimate $\widehat{Q}_p$ which is the $\lfloor pS \rfloor$-th order statistic in the buffer. Note that only the norms of previous gradients are stored in the buffer, limiting the memory overhead to $\mathcal{O}(S)$. The computational cost of $\widehat{Q}_p$ can also be kept to $\mathcal{O}(S)$ per iteration thanks to a bookkeeping procedure (see Appendix B).

Note that, since Algorithm 2 uses the corrupted samples $G(\theta_t, \zeta_t)$ rather than the true ones $\widetilde{G}(\theta_t, \zeta_t)$ to estimate the quantiles, a more conservative upperbound of roughly $p \leq 1 - 2\eta$ should be respected when an estimate of $\eta$ is available. Otherwise, one may default to $p = 1/2$ as an initial guess and adapt based on performance. In practice, our experiments show that relatively low values within $p \in [0.1, 0.2]$ are best for strongly convex objectives while higher values are affordable in other cases. See Appendix B for more details.

We implement this procedure for a few tasks and compare its performance with relevant baselines. We do not include a comparison with (Diakonikolas et al., 2022) whose procedure has no implementation we are aware of and is difficult to use in practice. Indeed, the algorithm in question heavily depends on several problem parameters and involves a filtering procedure which requires multiple passes on large data mini-batches making it impractical for the streaming setting. Moreover, a number of special methods are required to mitigate the costs of the matrix operations needed in the original procedure making the algorithm's implementation even more involved.

Our experiments on synthetic data consider an infinite horizon, dimension $d = 128$, and a constant step size for all methods.

**Linear regression.** We consider least-squares linear regression and compare RQC-SGD with Huber's estimator (Huber, 1973) and clipped SGD (designated as CClip($\lambda$)) with three clipping levels $\lambda \sigma_{\max} \sqrt{d}$ for $\lambda \in \{0.8, 1.0, 1.2\}$ where $\sigma_{\max}$ is a fixed data scaling factor. These thresholds provide a rough estimate of the gradient norm near the optimum $\theta^\star$. We generate covariates $X$ and labels $Y$ both heavy-tailed and corrupted. Corruption in the data stream is generated according to Assumption 3 with outliers represented either by aberrant values or *fake* samples $Y = X^\top \theta_{\text{fake}} + \epsilon$ using a false parameter $\theta_{\text{fake}}$, see Appendix B for further details on data generation and fine tuning of the Huber parameter. All methods are run with constant step size and averaged results over 100 runs are displayed on Figure 1 (top row).

As anticipated, Huber's loss function is not robust to corrupted covariates. In contrast, using gradient clipping allows convergence to meaningful estimates. Although this holds true for a constant threshold, Figure 1 shows it may considerably slow the convergence if started away from the optimum. In addition, the clipping level also affects the final estimation precision and requires tuning. Both of the previous issues are well addressed by RQC-SGD whose adaptive clipping level allows fast progress of the optimization and accurate convergence towards a small neighborhood of the optimum.

**Logistic regression.** We test the same methods on logistic regression. Huber's baseline is represented by the modified Huber loss (also known as quadratic SVM (Zhang, 2004)). We generate data similarly to the previous task except for the labels which follow $Y \sim \text{Bernoulli}(\sigma(X^\top \theta^\star))$ with $\sigma$ the sigmoid function. Corrupted labels are either uninformative, flipped or obtained with a fake $\theta_{\text{fake}}$ (see details in Appendix B). Results are displayed on the bottom row of Figure 1.

As previously, Huber's estimator performs poorly with corruption. However, constant clipping appears to be better suited when the gradient is bounded, so that the optimization is less affected by its underestimation. We observe, nonetheless, that a higher clipping level may lead to poor convergence properties, even at a low corruption rate. Note also that the constant levels we use are based on prior knowledge about the data distribution and would have to be fine tuned in practice. Meanwhile, the latter issue is well addressed by quantile clipping. Finally, notice that no algorithm truly approaches the true solution for this task. This reflects the difficulty of improving upon Proposition 3 which only states convergence to a neighborhood where the objective gradient is comparable to the estimation error in magnitude.

**Classification with shallow networks.** Finally, we evaluate the performance on the task of training a single hidden layer neural network classifier on some real datasets which corresponds to a non-convex optimization problem. To handle multiclass data, we use the cross entropy loss and replace Huber's baseline with plain SGD for simplicity. We define constant clipping baselines using thresholds given by the quantiles of order $p = 0.25, 0.5$, and $0.75$ of the norms of a batch of gradients at the beginning of the optimisation. Due to the greater sensitivity to corruption observed in this case, we set $\eta = 0.02$ and use $p = 0.9$ for RQC-SGD.

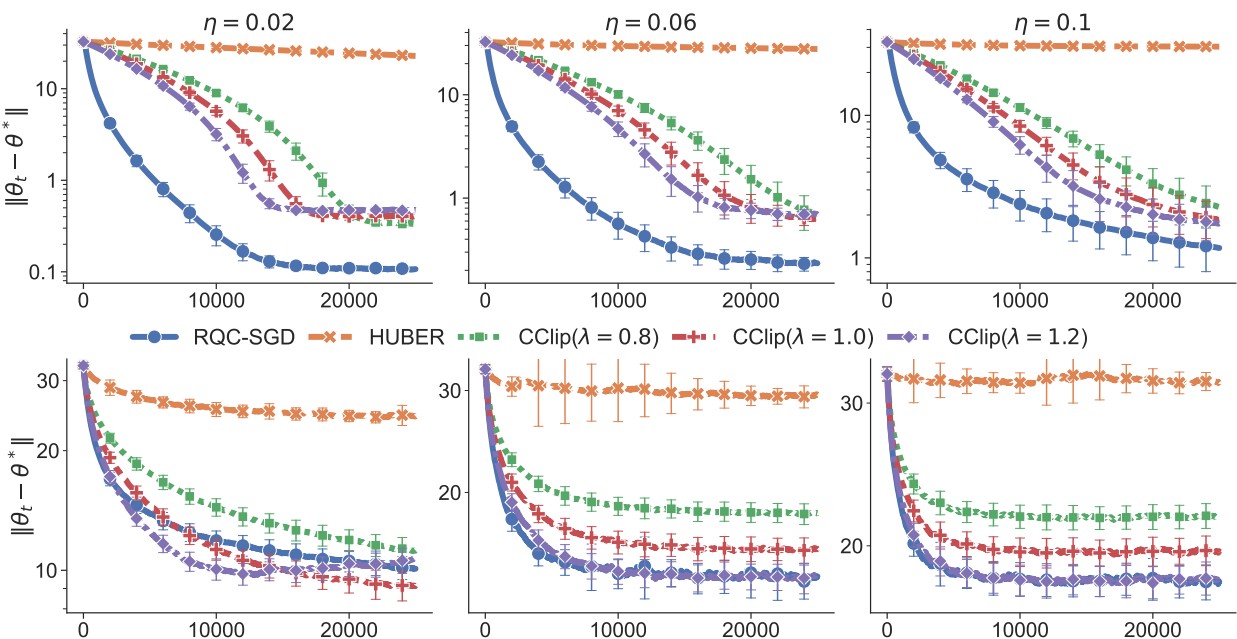

Figure 1: Evolution of $\|\theta_t - \theta^\star\|$ on the tasks of linear regression (top row) and logistic regression (bottom row) averaged over 100 runs at increasing corruption levels (error bars represent half the standard deviation). Estimators based on Huber's loss are strongly affected by data corruption. SGD with constant clipping thresholds is robust but slow to converge for linear regression and requires tuning for better final precision. RQC-SGD combines fast convergence with good final precision thanks to its adaptive clipping strategy.

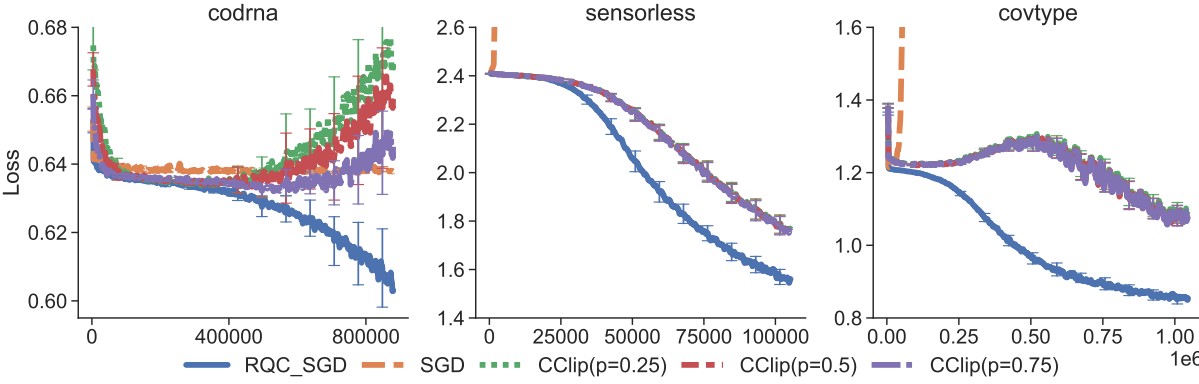

Figure 2: Evolution of the test loss ($y$-axis) against iteration $t$ ($x$-axis) for the training of a single hidden layer network on different real world classification datasets (average over 20 runs). We observe more consistent and stable objective decrease for RQC-SGD whereas constant clipping baselines are slower and may fail to converge.

We train all methods with one sample per iteration using equal step sizes and evaluate them through the test loss. We provide further results and experimental details in Appendix B. Results are displayed on Figure 2.

Unsurprisingly, standard SGD is not robust to corrupted samples and, while using a constant clipping level helps keep the optimisation on track, the experiments show that careful tuning may sometimes be necessary to prevent divergence. On the other hand, the adaptive clipping levels used by RQC-SGD allow to make the iteration faster *and* more resilient to corruption. This leads to an optimization path with a more consistent

decrease of the objective. Moreover, we also observe that RQC-SGD allows for a better control of the asymptotic variance of the optimized parameter compared to constant clipping.

## 6 Conclusion

We introduced a new clipping strategy for SGD and proved that it defines a stochastic optimization procedure which is robust to both heavy tails and outliers in the data stream. We also provided an efficient rolling quantile procedure to implement it and demonstrated its performance through numerical experiments on synthetic and real data. Future research directions include improving the dimension dependence in our bounds, possibly by using sample rejection rules or by considering stochastic mirror descent (Nemirovskij & Yudin, 1983; Beck & Teboulle, 2003) clipped with respect to a non Euclidean norm. This may also procure robustness to higher corruption rates. Another interesting research track is the precise quantification of the geometric convergence speed of the Markov chain generated by constant step size SGD on a strongly convex objective.

### Acknowledgments

This research is supported by the Agence Nationale de la Recherche as part of the "Investissements d'avenir" program (reference ANR-19-P3IA-0001; PRAIRIE 3IA Institute).

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

# Supplementary Material
## Robust Stochastic Optimization via Gradient Quantile Clipping

## A    Additional experimental results

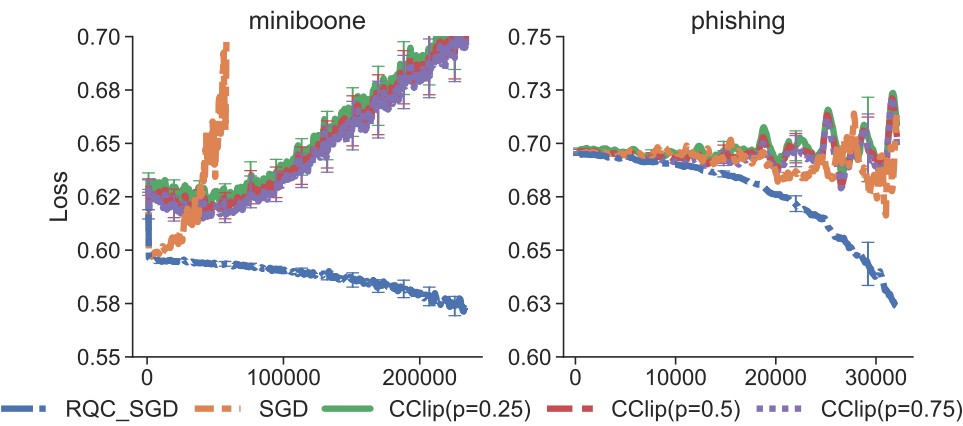

Figure 3: Evolution of the test loss ($y$-axis) against iteration $t$ ($x$-axis) for the training of a single hidden layer network on additional real world classification datasets (average over 20 runs).

**Classification with shallow networks.**   We performed the same experiment using two additional datasets. The results are displayed on Figure 3 and corroborate our statements in the main paper.

**Expectation estimation.**    We estimate the expectation of a random vector $X$ by minimizing the objective $\mathcal{L}(\theta) = \frac{1}{2}\|\theta - \theta^\star\|^2$ with $\theta^\star = \mathbb{E}[X]$ using a stream of both corrupted and heavy-tailed samples, see Appendix B for details. We run RQC-SGD (Algorithm 2) and compare it to an online version of geometric and coordinate-wise Median-Of-Means (GMOM and CMOM (Cardot et al., 2017; 2013)) which use block sample means to minimize an $L_1$ objective (see Appendix B). Although these estimators are a priori not robust to $\eta$-corruption, we ensure that their estimates are meaningful by limiting $\eta$ to 4% and using blocks of 10 samples. Thus, blocks are corrupted with probability $< 1/2$ so that the majority contains only true samples. Figure 4 displays the evolution of $\|\theta_t - \theta^\star\|$ for each method averaged over 100 runs for increasing $\eta$ and constant step size. We also display a single run for $\eta = 0.04$. We observe that RQC-SGD is only weakly affected by the increasing corruption whereas the performance of GMOM and CMOM quickly degrades with $\eta$, leading to unstable estimates.

## B    Experimental details

As previously mentioned, the dimension is set to $d = 128$ in our experiments with synthetic data. We also set $\sigma_{\min} = 1$ and $\sigma_{\max} = 5$ as minimum and maximum scaling factors. For all tasks and algorithms, the optimization starts from $\theta_0 = 0$.

**Bookkeeping in RQC-SGD**   The buffer in Algorithm 2 stores values in sorted order along with their "ages". The most recent and oldest values have ages 0 and $S - 1$ respectively. At each iteration, a new gradient is received, all ages are incremented and the oldest value is replaced by the new one with age 0. The latter is then sorted using one iteration of insertion sort. The estimate $\widehat{Q}_p$ is retrieved at each iteration as the value at position $\lfloor pS \rfloor$.

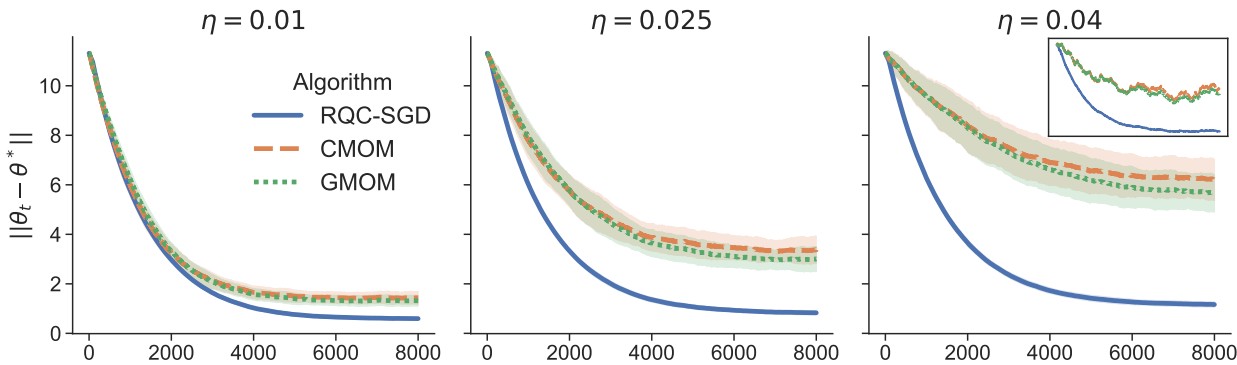

Figure 4: Evolution of $\|\theta_t - \theta^\star\|$ ($y$-axis) against iteration $t$ ($x$-axis) for the expectation estimation task, averaged over 100 runs at different corruption levels $\eta$ (bands widths correspond to the standard deviation of the 100 runs). For $\eta = 0.04$, the evolution on a single run is also displayed. We observe good performance for RQC-SGD for increasing $\eta$ while CMOM and GMOM are more sensitive.

## B.1 Mean estimation

**Data generation** We compute a matrix $\Sigma = (AA^\top + A^\top A)/2$ where $A \in \mathbb{R}^{d \times d}$ is a random matrix with i.i.d centered Gaussian entries with variance $1/d$ sampled once and for all. We generate true samples as $X = \mathbf{1} + \Sigma V$ where $V$ is a vector of i.i.d symmetrized Pareto random variables with parameter 2 and $\mathbf{1} \in \mathbb{R}^d$ denotes the vector with all entries equal to 1.

We draw corrupted samples as $\check{X} = 10\check{V} - 100 \times \mathbf{1}$ where $\check{V}$ is a vector of i.i.d symmetrized Pareto variables with parameter 1.5. We use step size $\beta = 10^{-3}$.

**GMOM and CMOM** The geometric and coordinatewise Median-Of-Means estimators (GMOM and CMOM) optimize the following objectives respectively:

$$\mathbb{E}\|\theta - \overline{X}_{N_b}\|_2 \quad \text{and} \quad \mathbb{E}\|\theta - \overline{X}_{N_b}\|_1,$$

where $\overline{X}_{N_b}$ is the average of $N_b$ independent copies of $X$. The block size is set to $N_b = 10$ in the whole experiment. The above objectives are optimized by computing samples of $\overline{X}_{N_b}$ in a streaming fashion so that one step is made for each $N_b$ samples. In order to compensate for this inefficiency we multiply the step size by $N_b$ for both GMOM and CMOM. For GMOM, we additionally multiply the step size by $\sqrt{d}$ in order to compensate the normalization included in the gradient formula.

**RQC-SGD** For mean estimation, we implement RQC-SGD (Algorithm 2) with buffer size $S = 100, p = 0.2$ and $\tau_{\text{unif}} = 10$.

## B.2 Linear regression

**Data generation** We choose the true parameter $\theta^\star$ by independently sampling its coordinate uniformly in the interval $[-5, +5]$. The true covariates are sampled as $X = \Sigma V$ where $\Sigma$ is a diagonal matrix with entries sampled uniformly in the interval $[\sigma_{\min}, \sigma_{\max}]$ (once and for all) and $V$ is a vector of i.i.d symmetrized Pareto random variables with parameter 2. The labels are sampled as $Y = X^\top \theta^\star + \epsilon$ where $\epsilon$ is a symmetrized Pareto random variable with parameter 2.

The corrupted samples are obtained according to one of the following possibilities with equal probability:

- $X = 1000(\max_i \Sigma_{ii})v + W$ where $v$ is a fixed unit vector and $W$ is a standard Gaussian vector and $Y \sim \text{Bernoulli}(1/2)$.

- $X = 1000(\max_i \Sigma_{ii})V$ with $V$ a unit norm random vector with uniform distribution and $Y = 1000(Z + U)$ where $Z$ is a random sign and $U$ is uniform over $[-1/5, 1/5]$.

- $X = 10V$ with $V$ a random vector with i.i.d entries following a symmetrized Log-normal distribution and $Y = X^\top \theta_{\text{fake}} + \epsilon$ with $\theta_{\text{fake}}$ a fake parameter drawn similarly to $\theta^\star$ once and for all and $\epsilon$ a standard Gaussian variable.

We use step size $\beta = 10^{-3}$.

**Huber parameter**   In order to tune the parameter $\delta$ of Huber's loss function, we proceed as follows:

- For each corruption level $\eta$, we consider 10 candidate values $\delta_j = 10^{j/2-5}$ for $0 \leq j < 10$.

- For each candidate $\delta_j$, we train 250 estimators $\left(\widehat{\theta}^{(i)}_{\delta_j}\right)_{i \in [\![250]\!]}$ using 1000 samples each.

- We choose $\widehat{j}$ for which the average $\frac{1}{250} \sum_{i \in [\![250]\!]} \left\| \widehat{\theta}^{(i)}_{\delta_j} - \theta^\star \right\|$ is minimal and use $\widehat{\delta} = \delta_{\widehat{j}}$ as parameter.

**RQC-SGD**   For linear regression, we run RQC-SGD with buffer size $S = 100$ and $\tau_{\text{unif}} = 10$. The quantile value was set to $p = 0.1$ for $\eta \in \{0.02, 0.06\}$ and $p = 0.05$ for $\eta = 0.1$.

### B.3   Logistic regression

**Data generation**   The true parameter $\theta^\star$ and covariates $X$ are chosen similarly to linear regression. Given $X$, the label $Y$ is set to $+1$ with probability $\sigma(X^\top \theta^\star)$ where $\sigma$ is the sigmoid function $\sigma(x) = (1 + e^{-x})^{-1}$ and to $-1$ otherwise.

The corrupted covariates are determined similarly to linear regression while the labels are set as follows in each respective case:

- $Y$ is set to $+1$ or $-1$ with equal probability.

- $Y = -\text{sign}(X^\top \theta^\star)$.

- $Y = \text{sign}(X^\top \theta_{\text{fake}})$ with $\theta_{\text{fake}}$ a fake parameter drawn similarly to $\theta^\star$ once and for all.

We use step size $\beta = 6 \times 10^{-3}$.

**Huber parameter**   The same procedure is used to tune the parameter of the modified Huber loss as for linear regression.

**RQC-SGD**   For logistic regression, we run RQC-SGD with buffer size $S = 100$ and $\tau_{\text{unif}} = 10$. The quantile value was set to $p = 1 - \eta - 0.1$ for $\eta = 0.02$ and $p = 1 - \eta - 0.05$ otherwise.

### B.4   Single hidden layer neural network classifier

We train a single hidden layer neural network classifier with 100 hidden neurones for all datasets. We use one sample per iteration and step size $\beta = 10^{-2}$ for all methods.

As previously, RQC-SGD is run with buffer size $S = 100$ and $\tau_{\text{unif}} = 10$. The quantile value was set to $p = 0.9$. We compute the gradient norms over a batch of samples of size $S$ at the beginning of the optimization and use the quantiles of order $p = 0.25, 0.5$ and $0.75$ as the clipping level for the constant clipping baselines.

**Data**   We used publicly available datasets for our experiments. We provide details about their characteristics and sources in Table 1.

We use a 10% share of each dataset as a test set in order to compute the test loss plotted in Figures 2 and 3. We also ensure the test set contains at least 5000 elements. Optimization is run using the remaining train set which is corrupted as specified next. The results are averaged over 20 runs for each datasets.

| Dataset | # Samples | # Features | # Classes | Source |
|---------|-----------|------------|-----------|--------|
| Codrna (Uzilov et al., 2006) | 488,565 | 8 | 2 | OpenML |
| Sensorless (Bator, 2015) | 58,509 | 48 | 11 | UCI |
| Covtype (Blackard, 1998) | 581,012 | 52 | 7 | scikit-learn |
| Miniboone (Roe, 2010) | 130,065 | 50 | 2 | UCI |
| Phishing (Hannousse & Yahiouche, 2020) | 11,430 | 87 | 2 | Kaggle |

Table 1: Main characteristics of the data sets used in experiments, including number of samples, number of features, number of classes and sources.

**Data corruption**  We corrupt train data samples at each iteration uniformly at random with probability $\eta = 0.02$.

Although we run the optimization using one sample per iteration, the datasets we use are available offline so that we have a data matrix denoted $\boldsymbol{X} \in \mathbb{R}^{n \times (d+1)}$ whose last column represents the labels. This corresponds to $n$ samples and $d$ features.

For each feature $j \in \llbracket d \rrbracket$, we compute $\widehat{\mu}_j$ and $\widehat{\sigma}_j$ the empirical mean and standard deviation respectively. We also sample a random unit vector $u$ of size $d$ and introduce corruption as follows:

- For the label column, we introduce corruption by changing the value uniformly at random among the other possible modalities.

- For features, we introduce corruption by replacing the original values with one of the following possibilities with equal probability:
  - a vector $\xi$ sampled coordinatewise according to $\xi_j = r_j + 1000 \times \widehat{\sigma}_j \nu$ where $r_j$ is a value randomly picked in the column $\boldsymbol{X}_{\bullet,j}$ and $\nu$ is a sample from the Student distribution with 2.1 degrees of freedom.
  - a vector $\xi$ sampled coordinatewise according to $\xi_j = \widehat{\mu}_j + 1000 \times \widehat{\sigma}_j u_j + z$ where $z$ is a standard Gaussian.
  - a vector $\xi$ sampled according to $\xi = \widehat{\mu} + 1000 \times \widehat{\sigma} \otimes w$ where $w$ is a uniformly sampled unit vector.

## C   Geometric convergence speed and relation to step size

The geometric Markov chain convergence stated in Theorem 1 occurs at a speed determined by the contraction factor $\rho$ which mainly depends on the step size $\beta$ and quantile $p$ defining the iteration. Therefore, an explicit formulation of this dependency is necessary to precisely quantify the convergence speed. This question is lightly touched upon in (Yu et al., 2021) whose Proposition 2.1 is an analogous SGD ergodicity result. Like Theorem 1, the latter relies on the Markov chain theory presented in (Meyn & Tweedie, 1993). It is argued in (Yu et al., 2021) that a vanishing step size $\beta \to 0$ causes $\rho$ to be close to one, leading to slow convergence but with smaller bias. However, these considerations remain asymptotic and do not quite address the convergence speed issue.

More generally, the precise estimation of the factor $\rho$ goes back to the evaluation of the convergence speed of a Markov chain satisfying a geometric drift property. Near optimal results exist for chains with particular properties such as stochastic order (Lund et al., 1996; Roberts & Tweedie, 2000), reversibility (Jerison, 2019) or special assumptions on the renewal distribution (Berenhaut & Lund, 2001). Unfortunately, such properties do not hold for SGD. Let $(\theta_t)_{t \geq 0}$ be a Markov chain satisfying the drift property:

$$\Delta V(\theta) \leq \begin{cases} (1 - \lambda)V(\theta) & \text{for} \quad \theta \notin \mathcal{C} \\ b & \text{for} \quad \theta \in \mathcal{C} \end{cases}$$

with $\lambda \in (0, 1), b < +\infty, V$ a real function such that $V(\theta) \geq 1$ for all $\theta$ and $\mathcal{C}$ a (bounded) small set (see (Meyn & Tweedie, 1993, Chapter 5)). Then, based on the available literature (Baxendale, 2005; Bednorz, 2013),

$(\theta_t)_{t \geq 0}$ converges as in Theorem 1 with $\rho \approx 1 - \lambda^3$, the latter estimation being unimprovable without further information on $(\theta_t)_{t \geq 0}$ (see the discussion following (Baxendale, 2005, Theorem 3.2)). For the specific setting of Theorem 1 (and more generally for SGD by setting $p = 1$), this only yields an excessively pessimistic estimate

$$\rho \approx 1 - (p\beta\mu)^3 \tag{17}$$

whereas it is reasonable to conjecture that $\rho \approx 1 - p\beta\mu$. The suboptimality of (17) is felt in the uncorrupted case in Proposition 2 and Corollary 1 where one is tempted to set $\beta$ of order $1/T$, with $T$ the horizon, reducing the bias to $\mathcal{O}(1/\sqrt{T})$. However, this results in an unacceptable sample cost of order $T^3$ before convergence occurs. On the other hand, assuming the estimate $\rho \approx 1 - p\beta\mu$ holds, using a step size of order $\log(T)/T$ allows to combine fast convergence and near optimal statistical performance. Finally, note that in the corrupted case, the optimal statistical rate is $\mathcal{O}(\eta^{1-1/q})$ so that striking such a compromise is unnecessary.

# D Proofs

## D.1 Preliminary lemmas

**Lemma 1.** *Grant Assumptions 1 and 2. For any $\theta, \theta' \in \mathbb{R}^d$ and $\beta \leq \frac{2}{\mu+L}$ we have :*

$$\left\| \theta - \beta\nabla\mathcal{L}(\theta) - (\theta' - \beta\nabla\mathcal{L}(\theta')) \right\|^2 \leq (1-\beta\mu)^2\|\theta-\theta'\|^2 \tag{18}$$

*Proof.* For $\beta \leq \frac{2}{\mu+L}$, we have:

$$
\begin{aligned}
\left\| \theta - \beta\nabla\mathcal{L}(\theta) - (\theta' - \beta\nabla\mathcal{L}(\theta')) \right\|^2 & \\
&= \|\theta - \theta'\|^2 - 2\beta\langle\theta - \theta', \nabla\mathcal{L}(\theta) - \nabla\mathcal{L}(\theta')\rangle + \beta^2\|\nabla\mathcal{L}(\theta) - \nabla\mathcal{L}(\theta')\|^2 \\
&\leq (1 - \beta^2\mu L)\|\theta - \theta'\|^2 - \beta(2 - \beta(\mu+L))\langle\nabla\mathcal{L}(\theta) - \nabla\mathcal{L}(\theta'), \theta - \theta'\rangle \\
&\leq (1 - \beta^2\mu L)\|\theta - \theta'\|^2 - \beta(2 - \beta(\mu+L))\mu\|\theta - \theta'\|^2 \\
&= (1 - \beta^2\mu L - 2\beta\mu + \beta^2\mu(\mu+L))\|\theta - \theta'\|^2 \\
&= (1 - \beta\mu)^2\|\theta - \theta'\|^2,
\end{aligned}
$$

where we used the inequalities :

$$\left\| \nabla\mathcal{L}(\theta) - \nabla\mathcal{L}(\theta') \right\|^2 \leq (\mu+L)\langle\nabla\mathcal{L}(\theta) - \nabla\mathcal{L}(\theta'), \theta - \theta'\rangle - \mu L\|\theta - \theta'\|^2 \tag{19}$$

$$\mu\|\theta - \theta'\|^2 \leq \langle\nabla\mathcal{L}(\theta) - \nabla\mathcal{L}(\theta'), \theta - \theta'\rangle, \tag{20}$$

valid for all $\theta, \theta'$. Inequality (19) is stated, for example, in (Nesterov, 2014, Theoerem 2.1.12) (see also (Bubeck, 2015, Lemma 3.11) and (20) is just a characterization of strong convexity (see for instance (Nesterov, 2014, Theorem 2.1.9)). $\square$

Note that Lemma 1 is a simple generalization of the usual contraction property in strongly-convex optimization where $\theta'$ is usually taken as $\theta^\star$. In that case, one has $\nabla\mathcal{L}(\theta') = 0$. An example of such a result is given by (Bubeck, 2015, Theorem 3.10). A similar inequality to Lemma 1 was previously obtained in (Dieuleveut et al., 2020, Proposition 2) where convergence of SGD as Markov chain was studied.

In the sequel we will write gradient samples $G(\theta, \zeta)$ and $\widetilde{G}(\theta, \zeta)$ simply as $G(\theta)$ and $\widetilde{G}(\theta)$ respectively in order to lighten notation. The following lemma will be needed in the proof of Theorem 1.

**Lemma 2.** *Let Assumptions 4 and 5 hold. Let $\theta \in \mathbb{R}^d$ be fixed and let $\widetilde{G}(\theta) \sim \mathcal{D}_\mathcal{I}(\theta)$ be a non corrupted gradient sample. Choosing the clipping threshold as $\tau_\theta = Q_p(\|\widetilde{G}(\theta)\|)$ for some $p \in (0,1)$ and denoting $\alpha_\theta = \min\left(1, \frac{\tau_\theta}{\|G(\theta)\|}\right)$ the clipping factor and its average $\overline{\alpha}_\theta = \mathbb{E}\left[\alpha_\theta | \theta, G(\theta) = \widetilde{G}(\theta) \sim \mathcal{D}_\mathcal{I}(\theta)\right]$ we have:*

$$\left\| \mathbb{E}[\alpha_\theta \widetilde{G}(\theta)] - \overline{\alpha}_\theta\nabla\mathcal{L}(\theta) \right\| \leq (1-p)^{1-1/q}\left(A_q\|\theta - \theta^\star\| + B_q\right), \tag{21}$$

$$\tau_\theta \le \big\|\nabla\mathcal{L}(\theta)\big\| + Q_p(\|\varepsilon_\theta\|)$$
$$\le \big\|\nabla\mathcal{L}(\theta)\big\| + (1-p)^{-1/q}\big(A_q\|\theta-\theta^\star\| + B_q\big). \tag{22}$$

*If Assumption 5 holds with $q \ge 2$ then we also have*

$$\mathbb{E}\big\|\alpha_\theta\widetilde{G}(\theta) - \mathbb{E}[\alpha_\theta\widetilde{G}(\theta)]\big\|^2 \le \big(A_q\|\theta-\theta^\star\| + B_q\big)^2 + 5(1-p)\tau_\theta^2. \tag{23}$$

*Proof.* We condition on the event that the sample $G(\theta)$ is not corrupted i.e. $G(\theta) = \widetilde{G}(\theta) \sim \mathcal{D}_\mathcal{I}(\theta)$. Noticing that $\mathbf{1}_{\|\widetilde{G}(\theta)\|\le\tau_\theta} = 1 - \mathbf{1}_{\|\widetilde{G}(\theta)\|>\tau_\theta}$ and using the equality $\mathbb{E}[\widetilde{G}(\theta)] = \nabla\mathcal{L}(\theta)$, we find :

$$\mathbb{E}[\alpha_\theta\widetilde{G}(\theta)] - \overline{\alpha}_\theta\nabla\mathcal{L}(\theta) = \mathbb{E}\big[(\alpha_\theta - \overline{\alpha}_\theta)\big(\widetilde{G}(\theta) - \nabla\mathcal{L}(\theta)\big)\big]$$
$$= \mathbb{E}\big[(1-\overline{\alpha}_\theta)\big(\widetilde{G}(\theta) - \nabla\mathcal{L}(\theta)\big)\mathbf{1}_{\|\widetilde{G}(\theta)\|\le\tau_\theta}\big] + \mathbb{E}\big[(\alpha_\theta - \overline{\alpha}_\theta)\big(\widetilde{G}(\theta) - \nabla\mathcal{L}(\theta)\big)\mathbf{1}_{\|\widetilde{G}(\theta)\|>\tau_\theta}\big]$$
$$= (\overline{\alpha}_\theta - 1)\mathbb{E}\big[\big(\widetilde{G}(\theta) - \nabla\mathcal{L}(\theta)\big)\mathbf{1}_{\|\widetilde{G}(\theta)\|>\tau_\theta}\big] - \overline{\alpha}_\theta\mathbb{E}\big[\big(\widetilde{G}(\theta) - \nabla\mathcal{L}(\theta)\big)\mathbf{1}_{\|\widetilde{G}(\theta)\|>\tau_\theta}\big]$$
$$+ \mathbb{E}\big[\big(\tau_\theta/\|\widetilde{G}(\theta)\|\big)\big(\widetilde{G}(\theta) - \nabla\mathcal{L}(\theta)\big)\mathbf{1}_{\|\widetilde{G}(\theta)\|>\tau_\theta}\big]$$
$$= -\mathbb{E}\big[\big(1 - \tau_\theta/\|\widetilde{G}(\theta)\|\big)\big(\widetilde{G}(\theta) - \nabla\mathcal{L}(\theta)\big)\mathbf{1}_{\|\widetilde{G}(\theta)\|>\tau_\theta}\big].$$

Using our choice of $\tau_\theta$ and Hölder's inequality, we find :

$$\big\|\mathbb{E}[\alpha_\theta\widetilde{G}(\theta)] - \overline{\alpha}_\theta\nabla\mathcal{L}(\theta)\big\| \le \mathbb{E}\big[\mathbf{1}_{\|\widetilde{G}(\theta)\|>\tau_\theta}\big|1 - \tau_\theta/\|\widetilde{G}(\theta)\|\big|\big\|\widetilde{G}(\theta) - \nabla\mathcal{L}(\theta)\big\|\big]$$
$$\le \mathbb{E}\big[\mathbf{1}_{\|\widetilde{G}(\theta)\|>\tau_\theta}\big\|\widetilde{G}(\theta) - \nabla\mathcal{L}(\theta)\big\|\big]$$
$$\le (1-p)^{1-1/q}\mathbb{E}\big[\big\|\widetilde{G}(\theta) - \nabla\mathcal{L}(\theta)\big\|^q\big]^{1/q}$$
$$\le (1-p)^{1-1/q}\big(A_q\|\theta-\theta^\star\| + B_q\big),$$

where the second step corresponds to the inequality $\big|1 - \tau_\theta/\|\widetilde{G}(\theta)\|\big| \le 1$ under the event $\{\|\widetilde{G}(\theta)\| > \tau_\theta\}$. Note also that we used Assumption 5 since $\theta$ is fixed. Inequality (21) is now proven. To show (22), we first write the inequality:

$$\tau_\theta = Q_p(\|\widetilde{G}(\theta)\|) = Q_p(\|\nabla\mathcal{L}(\theta) + \varepsilon_\theta\|) \le \|\nabla\mathcal{L}(\theta)\| + Q_p(\|\varepsilon_\theta\|),$$

which holds since $\|\widetilde{G}(\theta)\|$ is a positive random variable. Further, using Assumption 5, we have:

$$1 - p = \mathbb{P}\big(\|\varepsilon_\theta\| > Q_p(\|\varepsilon_\theta\|)\big) \le \frac{\mathbb{E}[\|\varepsilon_\theta\|^q]}{Q_p(\|\varepsilon_\theta\|)^q} \le \Big(\frac{A_q\|\theta-\theta^\star\| + B_q}{Q_p(\|\varepsilon_\theta\|)}\Big)^q.$$

It only remains to take the $q$-th root and plug the obtained bound on $Q_p(\|\varepsilon_\theta\|)$ back above to obtain (22).

To show (23), we define the event $\mathcal{E} = \{\|\widetilde{G}(\theta)\| \le \tau_\theta\}$ and denote $\overline{\mathcal{E}}$ its complement such that $\mathbb{P}(\mathcal{E}) = p = 1 - \mathbb{P}(\overline{\mathcal{E}})$. We write

$$\mathbb{E}\big\|\alpha_\theta\widetilde{G}(\theta) - \mathbb{E}[\alpha_\theta\widetilde{G}(\theta)]\big\|^2 = p\mathbb{E}\big[\big\|\alpha_\theta\widetilde{G}(\theta) - \mathbb{E}[\alpha_\theta\widetilde{G}(\theta)]\big\|^2|\mathcal{E}\big]$$
$$+ (1-p)\mathbb{E}\big[\big\|\alpha_\theta\widetilde{G}(\theta) - \mathbb{E}[\alpha_\theta\widetilde{G}(\theta)]\big\|^2|\overline{\mathcal{E}}\big]$$
$$\le p\mathbb{E}\big[\big\|\alpha_\theta\widetilde{G}(\theta) - \mathbb{E}[\alpha_\theta\widetilde{G}(\theta)]\big\|^2|\mathcal{E}\big] + 4(1-p)\tau_\theta^2$$
$$= p\mathbb{E}\big[\big\|\alpha_\theta\widetilde{G}(\theta) - \mathbb{E}[\alpha_\theta\widetilde{G}(\theta)|\mathcal{E}]\big\|^2|\mathcal{E}\big]$$
$$+ p\big\|\mathbb{E}[\alpha_\theta\widetilde{G}(\theta)|\mathcal{E}] - \mathbb{E}[\alpha_\theta\widetilde{G}(\theta)]\big\|^2 + 4(1-p)\tau_\theta^2$$
$$= p\mathbb{E}\big[\big\|\alpha_\theta\widetilde{G}(\theta) - \mathbb{E}[\alpha_\theta\widetilde{G}(\theta)|\mathcal{E}]\big\|^2|\mathcal{E}\big]$$
$$+ p\big\|(1-p)\mathbb{E}[\alpha_\theta\widetilde{G}(\theta)|\mathcal{E}] - (1-p)\mathbb{E}[\alpha_\theta\widetilde{G}(\theta)|\overline{\mathcal{E}}]\big\|^2 + 4(1-p)\tau_\theta^2$$
$$\le p\mathbb{E}\big[\big\|\alpha_\theta\widetilde{G}(\theta) - \mathbb{E}[\alpha_\theta\widetilde{G}(\theta)|\mathcal{E}]\big\|^2|\mathcal{E}\big] + 4p(1-p)^2\tau_\theta^2 + 4(1-p)\tau_\theta^2$$
$$\le p\mathbb{E}\big[\big\|\alpha_\theta\widetilde{G}(\theta) - \mathbb{E}[\alpha_\theta\widetilde{G}(\theta)|\mathcal{E}]\big\|^2|\mathcal{E}\big] + 5(1-p)\tau_\theta^2$$

where we used the identity $\mathbb{E}[\alpha_\theta \widetilde{G}(\theta)] = p\mathbb{E}[\alpha_\theta \widetilde{G}(\theta)|\mathcal{E}] + (1-p)\mathbb{E}[\alpha_\theta \widetilde{G}(\theta)|\overline{\mathcal{E}}]$ and the inequalities $\|\alpha_\theta \widetilde{G}(\theta)\| \leq \tau_\theta$ and $p(1-p) \leq 1/4$ which hold in all cases. In addition, we have

$$p\mathbb{E}\big[\big\|\alpha_\theta \widetilde{G}(\theta) - \mathbb{E}[\alpha_\theta \widetilde{G}(\theta)|\mathcal{E}]\big\|^2\big|\mathcal{E}\big] = p\mathbb{E}\big[\big\|\widetilde{G}(\theta) - \mathbb{E}[\widetilde{G}(\theta)|\mathcal{E}]\big\|^2\big|\mathcal{E}\big] \leq \mathbb{E}\big[\big\|\widetilde{G}(\theta) - \mathbb{E}[\widetilde{G}(\theta)]\big\|^2\big]$$
$$\leq \mathbb{E}\big[\big\|\widetilde{G}(\theta) - \nabla\mathcal{L}(\theta)\big\|^q\big]^{2/q} \leq \big(A_q\|\theta - \theta^\star\| + B_q\big)^2.$$

The first inequality is obtained by applying Lemma 3 to each coordinate of $\widetilde{G}(\theta)$ while conditioning on $\theta$. The second one uses Jensen's inequality and the third results from Assumption 5. $\qquad\square$

The statement of Lemma 2 appears to be novel and is enabled by the specific choice of the gradient norm quantile as clipping threshold.

**Lemma 3.** *Let $Y$ be a real random variable and $\mathcal{E}$ an event, then we have the inequality*

$$\mathbb{P}(\mathcal{E})\mathbb{E}[(Y - \mathbb{E}[Y|\mathcal{E}])^2|\mathcal{E}] \leq \mathbb{E}(Y - \mathbb{E}[Y])^2.$$

*Proof.* Define the conditional variance of a real random variable $Y$ w.r.t. another variable $Y$ as $\mathrm{Var}(Y|X) = \mathbb{E}[(Y - \mathbb{E}[Y|X])^2|X]$.

By Eve's law(see for instance (Blitzstein & Hwang, 2019, Theorem 9.5.4)) we have the identity

$$\mathrm{Var}(Y) = \mathbb{E}[\mathrm{Var}(Y|X)] + \mathrm{Var}(\mathbb{E}[Y|X]),$$

which entails the inequality $\mathrm{Var}(Y) \geq \mathbb{E}[\mathrm{Var}(Y|X)]$. Applying the latter with $X = \mathbf{1}_\mathcal{E}$ yields

$$\mathbb{E}(Y - \mathbb{E}[Y])^2 = \mathrm{Var}(Y) \geq \mathbb{P}(\mathcal{E})\mathbb{E}[(Y - \mathbb{E}[Y|\mathcal{E}])^2|\mathcal{E}] + (1 - \mathbb{P}(\mathcal{E}))\mathbb{E}[(Y - \mathbb{E}[Y|\overline{\mathcal{E}}])^2|\overline{\mathcal{E}}],$$

which implies the result. $\qquad\square$

We show the geometric ergodicity of the SGD Markov chain $(\theta_t)_{t\geq 0}$ by relying on (Meyn & Tweedie, 1993, Theorem 15.0.1). We will show that the following function :

$$V(\theta) := 1 + \|\theta - \theta^\star\|^2,$$

satisfies a *geometric drift* property. We define the action of the transition kernel $P_{\beta,p}$ on real integrable functions $f$ over $\mathbb{R}^d$ through:

$$P_{\beta,p}f(\theta) = \int f(\theta')P_{\beta,p}(\theta, d\theta') = \mathbb{E}\big[f(\theta - \alpha_\theta \beta G(\theta))\big].$$

We also define the variation operator:

$$\Delta f(\theta) := P_{\beta,p}f(\theta) - f(\theta).$$

In many of our proofs, we will make use of the following adjustable bound.

**Fact 1.** *For any real numbers $a, b$ and positive $\epsilon$, we have the inequality*

$$2ab \leq a^2\epsilon + b^2/\epsilon.$$

## D.2 Preliminary properties for Markov chains

In order to prove Theorem 1, we set up the formalism we need from (Meyn & Tweedie, 1993) through the following definitions.

**Definition 2** (Small set). *A set $\mathcal{C} \in \mathcal{B}(\mathbb{R}^d)$ is called a small set if there exists an $m > 0$, and a non-trivial measure $\nu$ on $\mathcal{B}(\mathbb{R}^d)$ such that for all $\theta \in \mathcal{C}$ and $B \in \mathcal{B}(\mathbb{R}^d)$,*

$$P_{\beta,p}^m(\theta, B) \geq \nu(B).$$

*When this is the case, we say that $\mathcal{C}$ is $(m, \nu)$-small.*

Minorization properties such as the one above are useful for proving the convergence of Markov chains. We define an analogous one via the notion of sampled chain.

**Definition 3** (Sampled chain). *Let $a$ be a probability measure on $\mathbb{Z}_+$ i.e. such that $a(n) \geq 0$ for $n \geq 0$ and $\sum_{n=0}^{\infty} a(n) = 1$. Then the transition kernel for the sampled Markov chain w.r.t. the distribution $a$ is*

$$K_a(\theta, A) = \sum_{n=0}^{\infty} P_{\beta,p}^n(\theta, A) a(n) \quad for \quad x \in \mathbb{R}^d, A \in \mathcal{B}(\mathbb{R}^d).$$

We define the notion of a *petite set* for a sampled Markov chain.

**Definition 4** (Petite set). *Let $a$ be a probability measure on $\mathbb{Z}_+$ defining a sampled chain. A set $\mathcal{C} \in \mathcal{B}(\mathbb{R}^d)$ is called a $(a, \nu)$-petite if*

$$K_a(\theta, B) \geq \nu(B),$$

*for all $\theta \in \mathcal{C}$ and $B \in \mathcal{B}(\mathbb{R}^d)$ where $\nu$ is a non-trivial measure on $\mathcal{B}(\mathbb{R}^d)$.*

Note that an $(m, \nu)$-small set is also $(a, \nu)$-petite with $a = \delta_m$. Finally, we define the norm of a measure w.r.t. a potential function.

**Definition 5** ($f$-norm). *Let $f : \mathbb{R}^d \to \mathbb{R}$ be a function such that $f \geq 1$ and let $\nu$ be a signed measure on $\mathbb{R}^d$. We define the $f$-norm of $\nu$ as*

$$\|\nu\|_f = \sup_{g:|g|\leq f} |\nu(g)| = \sup_{g:|g|\leq f} \left| \int g(\theta) \nu(d\theta) \right|.$$

*In particular, we have $\|\nu\|_f \geq \|\nu\|_{\mathrm{TV}}$.*

### D.3 Proof of Theorem 1

We will assume in this proof that $q \geq 2$. The case $q \in (1, 2)$ will be treated in Appendix D.5.

First, we define $\underline{\tau} = \inf_\theta \tau_\theta$. Note that Assumption 4 excludes the existence of $\theta$ such that $\widetilde{G}(\theta) = 0$ almost surely, therefore, we have $\underline{\tau} > 0$.

Thanks to Assumption 4 and conditioning on $\theta_t = \theta \in \mathbb{R}^d$ for $t \geq 0$, the distribution of $\theta_{t+1}$ has a strictly positive density at least on a ball of radius $\beta\underline{\tau}$ around $\theta_t$. This implies that the chain is aperiodic since $P_{\beta,p}(\theta_t, W_{\theta_t}) > 0$ for any neighborhood $W_{\theta_t}$ contained in the previous ball. Moreover, by induction, the distribution of $\theta_{t+m}$ has positive density at least on a ball of radius $m\beta\underline{\tau}$ around $\theta_t$. Thus for $m$ high enough we have $P_{\beta,p}^m(\theta_t, A) > 0$ for any set $A$ with non zero Lebesgue measure. It follows that the Markov chain is irreducible w.r.t. Lebesgue's measure and is thus $\psi$-irreducible (see (Meyn & Tweedie, 1993, Chapter 4)).

For fixed $\theta$, condition (9) implies $\beta < \frac{2}{\mu+L}$ and using Lemma 1 and denoting $\alpha_\theta$ and $\overline{\alpha}_\theta$ as in Lemma 2, we find:

$$
\begin{aligned}
P_{\beta,p}\|\theta - \theta^\star\|^2 &= \mathbb{E}\|\theta - \beta\alpha_\theta G(\theta) - \theta^\star\|^2 \\
&\leq \eta\mathbb{E}\big(\|\theta - \theta^\star\| + \beta\tau_\theta\big)^2 + (1-\eta)\mathbb{E}\big[\|\theta - \alpha_\theta\beta\widetilde{G}(\theta) - \theta^\star\|^2\big] \\
&\leq \eta\mathbb{E}\big(\|\theta - \theta^\star\| + \beta\tau_\theta\big)^2 + (1-\eta)\mathbb{E}\big[\|\theta - \overline{\alpha}_\theta\beta\nabla\mathcal{L}(\theta) - \theta^\star\|^2 - \\
&\quad 2\beta\langle\theta - \overline{\alpha}_\theta\beta\nabla\mathcal{L}(\theta) - \theta^\star, \alpha_\theta\widetilde{G}(\theta) - \overline{\alpha}_\theta\nabla\mathcal{L}(\theta)\rangle + \beta^2\|\alpha_\theta\widetilde{G}(\theta) - \overline{\alpha}_\theta\nabla\mathcal{L}(\theta)\|^2\big] \\
&\leq \eta\mathbb{E}\big(\|\theta - \theta^\star\| + \beta\tau_\theta\big)^2 + (1-\eta)\mathbb{E}\big[\big(\|\theta - \overline{\alpha}_\theta\beta\nabla\mathcal{L}(\theta) - \theta^\star\| \\
&\quad + \beta\|\mathbb{E}[\alpha_\theta\widetilde{G}(\theta)] - \overline{\alpha}_\theta\nabla\mathcal{L}(\theta)\|\big)^2 + \beta^2\big(A_q\|\theta - \theta^\star\| + B_q\big)^2 + 5\beta^2(1-p)\tau_\theta^2\big],
\end{aligned}
$$

where the last step uses that $\mathbb{E}\|\alpha_\theta\widetilde{G}(\theta) - \overline{\alpha}_\theta\nabla\mathcal{L}(\theta)\|^2 = \mathbb{E}\|\alpha_\theta\widetilde{G}(\theta) - \mathbb{E}[\alpha_\theta\widetilde{G}(\theta)]\|^2 + \|\mathbb{E}[\alpha_\theta\widetilde{G}(\theta)] - \overline{\alpha}_\theta\nabla\mathcal{L}(\theta)\|^2$ and Lemma 2.

Using Lemmas 1 and 2 and Assumption 5 and grouping the terms by powers of $\|\theta - \theta^\star\|$, we arrive at

$$
\begin{aligned}
P_{\beta,p}\|\theta - \theta^\star\|^2 &\leq \big(\eta + (1-\eta)(1 - \overline{\alpha}_\theta \beta\mu)^2\big)\mathbb{E}\|\theta - \theta^\star\|^2 + \\
&\quad 2\beta\mathbb{E}\big[\|\theta - \theta^\star\|\big(\eta\tau_\theta + (1-\eta)(1 - \overline{\alpha}_\theta\beta\mu)\big\|\mathbb{E}[\alpha_\theta\widetilde{G}(\theta)] - \overline{\alpha}_\theta\nabla\mathcal{L}(\theta)\big\|\big)\big] + \\
&\quad \beta^2\big(\eta\tau_\theta^2 + (1-\eta)\big(\big\|\mathbb{E}[\alpha_\theta\widetilde{G}(\theta)] - \overline{\alpha}_\theta\nabla\mathcal{L}(\theta)\big\|^2 + (A_q\|\theta - \theta^\star\| + B_q)^2 + 5(1-p)\tau_\theta^2\big)\big) \\
&\leq \mathbb{E}\big[\mathfrak{A}\|\theta - \theta^\star\|^2 + 2\beta\mathfrak{B}\|\theta - \theta^\star\|\big] + \beta^2\mathfrak{C} \\
&\leq (\mathfrak{A} + \beta\kappa)\mathbb{E}\big[\|\theta - \theta^\star\|^2\big] + \frac{\beta\mathfrak{B}^2}{\kappa} + \beta^2\mathfrak{C},
\end{aligned}
\tag{24}
$$

where we used Fact 1 and defined the quantities $\mathfrak{A}, \mathfrak{B}, \mathfrak{C}$ which may be bounded as follows

$$
\begin{aligned}
\mathfrak{A} &= \eta + (1-\eta)(1 - \overline{\alpha}_\theta\beta\mu)^2 + 2\beta(\eta(L + A_q(1-p)^{-1/q}) + (1-\eta)(1 - \overline{\alpha}_\theta\beta\mu)(1-p)^{-1/q}A_q) \\
&\quad + \beta^2((\eta + 5(1-p))(L + (1-p)^{-1/q}A_q)^2 + 4A_q^2) \\
&\leq 1 - 2\beta\big((1-\eta)\overline{\alpha}_\theta\mu - \eta(L + A_q(1-p)^{-1/q}) - (1-\eta)(1 - \overline{\alpha}_\theta\beta\mu)(1-p)^{1-1/q}A_q\big) + \\
&\quad \beta^2\big((1-\eta)(\overline{\alpha}_\theta\mu)^2 + 2(\eta + 5(1-p))\big(L + (1-p)^{-\frac{1}{q}}A_q\big)^2 + 4A_q^2\big) \\
&\leq 1 - 2\beta\big((1-\eta)p\mu - \eta L - (1-p)^{-\frac{1}{q}}A_q\big(1 - p(1-\eta)\big)\big) + \beta^2\big(\mu^2 + 24\eta L^2 + 28A_q^2\big) \\
&= 1 - 2\beta\kappa + \beta^2\big(\mu^2 + 24\eta L^2 + 28A_q^2\big), \\
\mathfrak{B} &= \big(\eta(1-p)^{-1/q} + (1-\eta)(1 - \overline{\alpha}_\theta\beta\mu)(1-p)^{1-1/q}\big)B_q \\
&\leq (1-p)^{-1/q}B_q(\eta + (1-p)) \\
\mathfrak{C} &= \big(4 + 2(\eta + 5(1-p))(1-p)^{-2/q}\big)B_q^2
\end{aligned}
$$

The above bounds use the simple properties $p \leq \overline{\alpha}_\theta \leq 1$, $0 \leq \eta \leq 1$, $1 - \overline{\alpha}_\theta\beta\mu \leq 1$ and $(a+b)^2 \leq 2a^2 + 2b^2$ for all $a, b$.

Thanks to our choice of $\beta$, we get that $\mathfrak{A} + \beta\kappa < 1$. It is now easy to check that $V(\theta) = 1 + \|\theta - \theta^\star\|^2$ satisfies the contraction

$$
P_{\beta,p}V(\theta) \leq \underbrace{(\mathfrak{A} + \beta\kappa)}_{=:\widetilde{\lambda}<1}V(\theta) + \underbrace{1 - (\mathfrak{A} + \beta\kappa) + \frac{\beta\mathfrak{B}^2}{\kappa} + \mathfrak{C}}_{=:\widetilde{b}}.
$$

We now define the set $\mathcal{C} = \big\{\theta \in \mathbb{R}^d, \, V(\theta) \leq 2\widetilde{b}/(1 - \widetilde{\lambda})\big\}$ for which we have:

$$
\Delta V(\theta) \leq -\frac{1 - \widetilde{\lambda}}{2}V(\theta) + \widetilde{b}\mathbf{1}_{\theta\in\mathcal{C}}.
\tag{25}
$$

For such $\mathcal{C}$, let $\Delta_\mathcal{C} = \text{diam}(\mathcal{C})$ be its diameter and set $m_\mathcal{C} = \left\lceil\frac{\Delta_\mathcal{C}}{\beta\underline{\tau}}\right\rceil$. As previously mentioned in the beginning of the proof, conditioning on $\theta_t = \theta$, the distribution of $\theta_{t+m}$ admits a positive density at least over a ball of radius $m\beta\underline{\tau}$ around $\theta$ i.e. there exists $h_\theta^{+m}(\omega) \geq 0$ satisfying

$$
h_\theta^{+m}(\omega) > 0 \quad \text{for} \quad \|\theta - \omega\| < m\beta\underline{\tau} \quad \text{and} \quad P_{\beta,p}^m(\theta, A) \geq \int_A h_\theta^{+m}(\omega)d\omega \quad \text{for} \quad A \in \mathcal{B}(\mathbb{R}^d).
$$

We then let $\underline{h}_\mathcal{C}(\omega) = \inf_{\theta\in\mathcal{C}} h_\theta^{+m_\mathcal{C}}(\omega)$ and define the measure $\nu_\mathcal{C}$ by:

$$
\nu_\mathcal{C}(A) = \int_{A\cap\mathcal{C}} \underline{h}_\mathcal{C}(\theta)d\theta.
$$

The above measure is non trivial since our choice of $m_\mathcal{C}$ ensures that $\underline{h}_\mathcal{C}$ defines a non zero density at least on $\mathcal{C}$. It follows that for all $\theta_0 \in \mathcal{C}$, we have the following minorization property:

$$
P_{\beta,p}^{m_\mathcal{C}}(\theta_0, A) \geq \nu_\mathcal{C}(A) \quad \text{for all} \quad A \in \mathcal{B}(\mathbb{R}^d),
\tag{26}
$$

which implies that the set $\mathcal{C}$ is $(m_{\mathcal{C}}, \nu_{\mathcal{C}})$-*small* and also $(\delta_{m_{\mathcal{C}}}, \nu_{\mathcal{C}})$-*petite* as a result. We have previously shown that the Markov chain $(\theta_t)$ is irreducible and aperiodic. Moreover, we have exhibited a petite set $\mathcal{C}$ (via (26)) towards which the geometric drift property (25) holds. Thus, condition (iii) of (Meyn & Tweedie, 1993, Theorem 15.0.1) is fulfilled.

By the latter result, it follows that the chain $(\theta_t)$ admits a unique invariant probability measure $\pi_{\beta,p}$ and there exist $r > 1$ and $M < \infty$ such that:

$$\sum_{t \geq 0} r^t \|P_{\beta,p}(\theta_0, \cdot) - \pi_{\beta,p}\|_V \leq MV(\theta_0). \tag{27}$$

Taking $\rho = r^{-1}$ and using that $\|\nu\|_V \geq \|\nu\|_{\mathrm{TV}}$ for any signed measure $\nu$ concludes the proof.

### D.4 Proof of Proposition 1

We consider the case $q \geq 2$. Since the distribution $\pi_{\beta,p}$ is invariant by the transition kernel $P_{\beta,p}$, we can deduce that for $\theta \sim \pi_{\beta,p}$, we have

$$\mathbb{E}\|\theta - \theta^\star\|^2 = \mathbb{E}\|\theta - \beta\alpha_\theta G(\theta) - \theta^\star\|^2.$$

From here, we can follow similar computations to those leading to (24) in the proof of Theorem 1. By setting $p = 1 - \eta$ and using that $q \geq 2$, we additionally have the bounds $\mathfrak{B} \leq 3\eta^{1-1/q}B_q$ and $\mathfrak{C} \leq 16B_q^2$.

Hence, we have that:

$$\mathbb{E}\|\theta - \theta^\star\|^2 \leq (\mathfrak{A} + \beta\kappa)\mathbb{E}\|\theta - \theta^\star\|^2 + \frac{\beta\mathfrak{B}^2}{\kappa} + \beta^2\mathfrak{C}$$

$$\implies \mathbb{E}\|\theta - \theta^\star\|^2 \leq \frac{\beta\mathfrak{B}^2}{\kappa(1 - \mathfrak{A} - \beta\kappa)} + \frac{\beta^2\mathfrak{C}}{1 - \mathfrak{A} - \beta\kappa}. \tag{28}$$

Note that (9) entails

$$1 - \mathfrak{A} - \beta\kappa \geq \beta\kappa - \beta^2(\mu^2 + 24\eta L^2 + 28A_q^2) \geq 3\beta\kappa/4. \tag{29}$$

Plugging this into (28) and using (10), we find

$$\mathbb{E}\|\theta - \theta^\star\|^2 \leq \frac{12\eta^{2-2/q}B_q^2}{\kappa^2} + \frac{64\beta B_q^2}{3\kappa} \leq \frac{34\eta^{2-2/q}B_q^2}{\kappa^2}$$

which implies the result.

### D.5 The case $q \in (1, 2)$

Similar results hold for the case $q \in (1, 2)$ but require a different proof given below.

**Proposition 4.** *Let Assumptions 1-5 hold with $q \in (1, 2)$ and let QC-SGD be run with quantile $p \in [\eta, 1-\eta]$. Assume that*

$$\kappa' := (1 - \eta)p\mu - q\eta L - q(\eta(1-p)^{-1/q} + (1-p)^{1-1/q})A_q > 0, \tag{30}$$

*and take a step-size satisfying*

$$\beta \leq \left(\frac{\kappa'}{86(L + A_q)^q}\right)^{\frac{1}{q-1}}. \tag{31}$$

*then the generated Markov chain $(\theta_t)_t$ converges geometrically to a unique invariant measure $\pi_{\beta,1-\eta}$ as in Theorem 1. In addition, for $p = 1 - \eta, \beta \leq \eta/\kappa'$ and $\theta \sim \pi_{\beta,1-\eta}$, we have*

$$\mathbb{E}\|\theta - \theta^\star\|^q \leq 128\left(\frac{\eta^{1-1/q}B_q}{\kappa'}\right)^q.$$

*Proof.* We first need to prove that convergence holds as stated in Theorem 1. We establish the properties of irreducibility, aperiodicity and exhibit a petite similarly as before. However, the demonstration of a geometric drift property such as (25) requires a different approach.

In this proof, we will use the following inequalities valid for all positive $x, y$ and $\varepsilon$ and $q \in (1, 2)$

$$(x + y)^q \leq 2^{q-1}(x^q + y^q), \tag{32}$$

$$(x + y)^q \leq x^q + qx^{q-1}y + y^q, \tag{33}$$

(a consequence of the inequality $(1 + a)^q \leq 1 + qa + a^q$ for $a > 0$),

$$(x + y)^{q-1} \leq x^{q-1} + y^{q-1}, \tag{34}$$

and

$$xy \leq \frac{(x\varepsilon)^q}{q} + \frac{q-1}{q}(y/\varepsilon)^{q/(q-1)} \tag{35}$$

which is a consequence of Young's inequality applied to the pair $x\varepsilon$ and $y/\varepsilon$ with exponent $q$ and its conjugate. We first write

$$P_{\beta,p}\|\theta - \theta^\star\|^q = \mathbb{E}\|\theta - \beta\alpha_\theta G(\theta) - \theta^\star\|^q \leq \eta\mathbb{E}(\|\theta - \theta^\star\| + \beta\tau_\theta)^q + (1 - \eta)\mathbb{E}\|\theta - \alpha_\theta\beta\widetilde{G}(\theta) - \theta^\star\|^q$$

Defining the notation $\mathbb{E}_\theta[\cdot] = \mathbb{E}[\cdot|\theta]$, we have

$$\mathbb{E}\|\theta - \alpha_\theta\beta\widetilde{G}(\theta) - \theta^\star\|^q = \mathbb{E}\big[\big(\|\theta - \overline{\alpha}_\theta\beta\nabla\mathcal{L}(\theta) - \theta^\star\|^2 -$$
$$2\beta\langle\theta - \overline{\alpha}_\theta\beta\nabla\mathcal{L}(\theta) - \theta^\star, \alpha_\theta\widetilde{G}(\theta) - \overline{\alpha}_\theta\nabla\mathcal{L}(\theta)\rangle + \beta^2\|\alpha_\theta\widetilde{G}(\theta) - \overline{\alpha}_\theta\nabla\mathcal{L}(\theta)\|^2\big)^{q/2}\big]$$
$$\leq \mathbb{E}\big[\big(\|\theta - \overline{\alpha}_\theta\beta\nabla\mathcal{L}(\theta) - \theta^\star\|^2 - 2\beta\langle\theta - \overline{\alpha}_\theta\beta\nabla\mathcal{L}(\theta) - \theta^\star, \alpha_\theta\widetilde{G}(\theta) - \overline{\alpha}_\theta\nabla\mathcal{L}(\theta)\rangle +$$
$$\beta^2(\|\alpha_\theta\widetilde{G}(\theta) - \mathbb{E}_\theta[\alpha_\theta\widetilde{G}(\theta)]\|^2 + 2\langle\alpha_\theta\widetilde{G}(\theta) - \mathbb{E}_\theta[\alpha_\theta\widetilde{G}(\theta)], \mathbb{E}_\theta[\alpha_\theta\widetilde{G}(\theta)] - \overline{\alpha}_\theta\nabla\mathcal{L}(\theta)\rangle +$$
$$\|\mathbb{E}_\theta[\alpha_\theta\widetilde{G}(\theta)] - \overline{\alpha}_\theta\nabla\mathcal{L}(\theta)\|^2)\big)^{q/2}\big]$$
$$\leq \mathbb{E}\big[\big|\|\theta - \overline{\alpha}_\theta\beta\nabla\mathcal{L}(\theta) - \theta^\star\|^2 - 2\beta\langle\theta - \overline{\alpha}_\theta\beta\nabla\mathcal{L}(\theta) - \theta^\star, \alpha_\theta\widetilde{G}(\theta) - \overline{\alpha}_\theta\nabla\mathcal{L}(\theta)\rangle +$$
$$\beta^2(2\langle\alpha_\theta\widetilde{G}(\theta) - \mathbb{E}_\theta[\alpha_\theta\widetilde{G}(\theta)], \mathbb{E}_\theta[\alpha_\theta\widetilde{G}(\theta)] - \overline{\alpha}_\theta\nabla\mathcal{L}(\theta)\rangle + \|\mathbb{E}_\theta[\alpha_\theta\widetilde{G}(\theta)] - \overline{\alpha}_\theta\nabla\mathcal{L}(\theta)\|^2)\big|^{q/2}\big]$$
$$+ \beta^q\mathbb{E}\|\alpha_\theta\widetilde{G}(\theta) - \mathbb{E}_\theta[\alpha_\theta\widetilde{G}(\theta)]\|^q$$
$$\leq \mathbb{E}\big[\big|\|\theta - \overline{\alpha}_\theta\beta\nabla\mathcal{L}(\theta) - \theta^\star\|^2 - 2\beta\langle\theta - \overline{\alpha}_\theta\beta\nabla\mathcal{L}(\theta) - \theta^\star, \mathbb{E}_\theta[\alpha_\theta\widetilde{G}(\theta)] - \overline{\alpha}_\theta\nabla\mathcal{L}(\theta)\rangle +$$
$$\beta^2\|\mathbb{E}_\theta[\alpha_\theta\widetilde{G}(\theta)] - \overline{\alpha}_\theta\nabla\mathcal{L}(\theta)\|^2\big|^{q/2}\big] + \beta^q\mathbb{E}\|\alpha_\theta\widetilde{G}(\theta) - \mathbb{E}_\theta[\alpha_\theta\widetilde{G}(\theta)]\|^q$$
$$\leq \mathbb{E}\big(\|\theta - \overline{\alpha}_\theta\beta\nabla\mathcal{L}(\theta) - \theta^\star\| + \beta\|\mathbb{E}_\theta[\alpha_\theta\widetilde{G}(\theta)] - \overline{\alpha}_\theta\nabla\mathcal{L}(\theta)\|\big)^q + \beta^q\mathbb{E}\|\alpha_\theta\widetilde{G}(\theta) - \mathbb{E}_\theta[\alpha_\theta\widetilde{G}(\theta)]\|^q,$$

where we used (34), conditioned on $\theta$, then used Jensen's inequality for $\mathbb{E}_\theta$ and a Cauchy-Schwarz inequality. We focus on the last term. Defining the event $\mathcal{E} = \{\|\widetilde{G}(\theta)\| \leq \tau_\theta\}$ such that $\mathbb{P}(\mathcal{E}) = p$, we have

$$\mathbb{E}\|\alpha_\theta\widetilde{G}(\theta) - \mathbb{E}_\theta[\alpha_\theta\widetilde{G}(\theta)]\|^q = (1 - p)\mathbb{E}[\|\alpha_\theta\widetilde{G}(\theta) - \mathbb{E}_\theta[\alpha_\theta\widetilde{G}(\theta)]\|^q|\overline{\mathcal{E}}]$$
$$+ p\mathbb{E}[\|\alpha_\theta\widetilde{G}(\theta) - \mathbb{E}_\theta[\alpha_\theta\widetilde{G}(\theta)]\|^q|\mathcal{E}]$$
$$\leq (1 - p)(2\tau_\theta)^q + p\mathbb{E}[\|\alpha_\theta\widetilde{G}(\theta) - \mathbb{E}_\theta[\alpha_\theta\widetilde{G}(\theta)]\|^q|\mathcal{E}].$$

In addition, using (34) twice, we find

$$\mathbb{E}[\|\alpha_\theta\widetilde{G}(\theta) - \mathbb{E}_\theta[\alpha_\theta\widetilde{G}(\theta)]\|^q|\mathcal{E}] \leq 2^{q-1}\big(\mathbb{E}[\|\alpha_\theta\widetilde{G}(\theta) - \overline{\alpha}_\theta\nabla\mathcal{L}(\theta)\|^q|\mathcal{E}] +$$
$$\|\overline{\alpha}_\theta\nabla\mathcal{L}(\theta) - \mathbb{E}_\theta[\alpha_\theta\widetilde{G}(\theta)]\|^q\big)$$
$$\leq 2^{q-1}\big(2^{q-1}\mathbb{E}[\|\widetilde{G}(\theta) - \nabla\mathcal{L}(\theta)\|^q|\mathcal{E}] + 2^{q-1}(1 - \overline{\alpha}_\theta)^q\|\nabla\mathcal{L}(\theta)\|^q +$$
$$\|\overline{\alpha}_\theta\nabla\mathcal{L}(\theta) - \mathbb{E}_\theta[\alpha_\theta\widetilde{G}(\theta)]\|^q\big).$$

Therefore, from the two previous displays and using Assumption 1, Lemma 2 and the inequality $p\mathbb{E}[\|\widetilde{G}(\theta) - \nabla\mathcal{L}(\theta)\|^q | \mathcal{E}] \leq \mathbb{E}[\|\widetilde{G}(\theta) - \nabla\mathcal{L}(\theta)\|^q]$, we get

$$\mathbb{E}\|\alpha_\theta\widetilde{G}(\theta) - \mathbb{E}_\theta[\alpha_\theta\widetilde{G}(\theta)]\|^q \leq (1-p)(2\tau_\theta)^q + 2^{2q-2}p(1-\overline{\alpha}_\theta)^q L^q\|\theta - \theta^\star\|^q$$
$$+ (2^{2q-2} + 2^{q-1}p(1-p)^{q-1})(A_q\|\theta - \theta^\star\| + B_q)^q.$$

We also have

$$(1-p)\tau_\theta^q \leq (1-p)((L + (1-p)^{-1/q}A_q)\|\theta - \theta^\star\| + (1-p)^{-1/q}B_q)^q$$
$$\leq ((L + A_q)\|\theta - \theta^\star\| + B_q)^q.$$

Plugging into the previous display and simplifying leads to

$$\mathbb{E}\|\alpha_\theta\widetilde{G}(\theta) - \mathbb{E}_\theta[\alpha_\theta\widetilde{G}(\theta)]\|^q \leq 2^{2q-2}((L + A_q)\|\theta - \theta^\star\| + B_q)^q + 2^{2q-2}p(1-\overline{\alpha}_\theta)^q(L\|\theta - \theta^\star\|)^q$$
$$+ (2^{2q-2} + 2^{q-1}p(1-p)^{q-1})(A_q\|\theta - \theta^\star\| + B_q)^q$$
$$\leq 2^{2q-1}((L + A_q)\|\theta - \theta^\star\| + B_q)^q + (2^{2q-2} + 2^{q-1}p(1-p)^{q-1})(A_q\|\theta - \theta^\star\| + B_q)^q$$
$$\leq 14((L + A_q)\|\theta - \theta^\star\| + B_q)^q. \tag{36}$$

Using these inequalities along with Lemma 2 to bound $\mathbb{E}\|\theta - \theta^\star\|^q$, we find that

$$P_{\beta,p}\|\theta - \theta^\star\|^q \leq \eta\mathbb{E}\big(\|\theta - \theta^\star\| + \beta\tau_\theta\big)^q + (1-\eta)\beta^q\mathbb{E}\|\alpha_\theta\widetilde{G}(\theta) - \mathbb{E}_\theta[\alpha_\theta\widetilde{G}(\theta)]\|^q$$
$$+ (1-\eta)\mathbb{E}\big(\|\theta - \overline{\alpha}_\theta\beta\nabla\mathcal{L}(\theta) - \theta^\star\| + \beta\|\mathbb{E}_\theta[\alpha_\theta\widetilde{G}(\theta)] - \overline{\alpha}_\theta\nabla\mathcal{L}(\theta)\|\big)^q$$
$$\leq \eta\mathbb{E}\big(\|\theta - \theta^\star\|^q + \beta^q\tau_\theta^q + q\beta\|\theta - \theta^\star\|^{q-1}\tau_\theta\big) + (1-\eta)\mathbb{E}\big((1 - \overline{\alpha}_\theta\beta\mu)^q\|\theta - \theta^\star\|^q$$
$$+ \beta^q\|\mathbb{E}_\theta[\alpha_\theta\widetilde{G}(\theta)] - \overline{\alpha}_\theta\nabla\mathcal{L}(\theta)\|^q + q\beta\|\theta - \theta^\star\|^{q-1}\|\mathbb{E}_\theta[\alpha_\theta\widetilde{G}(\theta)] - \overline{\alpha}_\theta\nabla\mathcal{L}(\theta)\|\big)$$
$$+ (1-\eta)\beta^q\mathbb{E}\|\alpha_\theta\widetilde{G}(\theta) - \mathbb{E}_\theta[\alpha_\theta\widetilde{G}(\theta)]\|^q$$
$$\leq \mathbb{E}\big[(1 - (1-\eta)\overline{\alpha}_\theta\beta\mu)\|\theta - \theta^\star\|^q\big] + q\beta\mathbb{E}\big[\|\theta - \theta^\star\|^{q-1}\big(\eta\tau_\theta$$
$$+ (1-\eta)\|\mathbb{E}_\theta[\alpha_\theta\widetilde{G}(\theta)] - \overline{\alpha}_\theta\nabla\mathcal{L}(\theta)\|\big) + \beta^q\big(\eta\tau_\theta^q$$
$$+ (1-\eta)\big(\|\mathbb{E}_\theta[\alpha_\theta\widetilde{G}(\theta)] - \overline{\alpha}_\theta\nabla\mathcal{L}(\theta)\|^q + \|\alpha_\theta\widetilde{G}(\theta) - \mathbb{E}_\theta[\alpha_\theta\widetilde{G}(\theta)]\|^q\big)\big)\big]$$
$$\leq \big(1 - \beta((1-\eta)p\mu - q\eta(L + (1-p)^{-1/q}A_q) - q(1-\eta)(1-p)^{1-1/q}A_q)$$
$$+ \beta^q 2^{q-1}(\eta(L + (1-p)^{-1/q}A_q)^q + (1-\eta)14(L + A_q)^q$$
$$+ (1-\eta)(1-p)^{q-1}A_q^q))\mathbb{E}\|\theta - \theta^\star\|^q + q\beta\mathbb{E}\|\theta - \theta^\star\|^{q-1}\big((1-\eta)(1-p)^{1-1/q}B_q$$
$$+ \eta(1-p)^{-1/q}B_q\big) + \beta^q 2^{q-1}\big(\eta(1-p)^{-1}B_q^q + (1-\eta)(1-p)^{q-1}B_q^q + 14B_q^q\big)$$
$$\leq \mathfrak{A}\mathbb{E}\|\theta - \theta^\star\|^q + q\beta\mathfrak{B}\mathbb{E}\|\theta - \theta^\star\|^{q-1} + \beta^q 2^{q-1}\mathfrak{C}^q \tag{37}$$

where the second inequality uses Lemma 1 and (33) twice, the third inequality rearranges according to powers of $\|\theta - \theta^\star\|$, the fourth one applies Lemma 2 and (36) and the last inequality defines the factors $\mathfrak{A}, \mathfrak{B}$ and $\mathfrak{C}$. Since $p \leq 1 - \eta$, these satisfy:

$$\mathfrak{A} \leq 1 - \beta\kappa' + \beta^q 2^{q-1}(16)(L + A_q)^q, \quad \mathfrak{B} \leq 2B_q \quad \text{and} \quad \mathfrak{C} \leq 2^{4/q}B_q$$

Applying (35), we have for all $\varepsilon > 0$ that

$$\mathfrak{B}\mathbb{E}\|\theta - \theta^\star\|^{q-1} \leq \frac{\varepsilon^{\frac{q}{q-1}}\mathbb{E}\|\theta - \theta^\star\|^q}{q/(q-1)} + \frac{\mathfrak{B}/\varepsilon)^q}{q} \leq \frac{\kappa'\mathbb{E}\|\theta - \theta^\star\|^q}{8q} + \frac{\mathfrak{B}^q}{q}\Big(\frac{\kappa'}{8(q-1)}\Big)^{1-q}, \tag{38}$$

where the second inequality corresponds to the choice $\varepsilon = \Big(\frac{\kappa'}{8(q-1)}\Big)^{\frac{q-1}{q}}$. Plugging back into (37) and using condition (31) ensures that $\mathfrak{A} + \beta\kappa'/8 < 1$. Defining $V(\theta) = 1 + \|\theta - \theta^\star\|^q$, we get that satisfies the contraction

$$P_{\beta,p}V(\theta) \leq \underbrace{(\mathfrak{A} + \beta\kappa'/8)}_{=:\widetilde{\lambda} < 1}V(\theta) + \underbrace{1 - (\mathfrak{A} + \beta\kappa'/8) + \beta\mathfrak{B}^q\Big(\frac{\kappa'}{8(q-1)}\Big)^{1-q} + \beta^q 2^{q-1}\mathfrak{C}^q}_{=:\widetilde{b}}.$$

From here, we use the same argument as the previous proof of Theorem 1 to deduce that the Markov chain converges to a unique invariant distribution $\pi_{\beta,p}$. Now, since convergence occurs to a limit distribution $\pi_{\beta,p}$, we have, as before, for $\theta \sim \pi_{\beta,p}$ the identity

$$\mathbb{E}\|\theta - \theta^\star\|^q = \mathbb{E}\|\theta - \beta\alpha_\theta G(\theta) - \theta^\star\|^q.$$

By setting $p = 1 - \eta$, we can show that in fact $\mathfrak{B} \leq 2\eta^{1-1/q} B_q$. Plugging back into (37) and using (38) and (31), we find

$$\mathbb{E}\|\theta - \theta^\star\|^q \leq \left(1 - (7/8)\beta\kappa' + 32\beta^q(L + A_q)^q\right)\mathbb{E}\|\theta - \theta^\star\|^q + \beta\left(2\eta^{1-1/q}B_q\right)^q\left(\frac{\kappa'}{8(q-1)}\right)^{1-q} + 32\beta^q B_q^q$$

$$\leq \left(1 - (3/8)\beta\kappa'\right)\mathbb{E}\|\theta - \theta^\star\|^q + \beta\left(2\eta^{1-1/q}B_q\right)^q\left(\frac{\kappa'}{8(q-1)}\right)^{1-q} + 32\beta^q B_q^q$$

Now using the condition $\beta \leq \eta/\kappa'$ and rearranging the inequality, we finally arrive at

$$\mathbb{E}\|\theta - \theta^\star\|^q \leq \frac{8}{3}\left(\frac{\eta^{1-1/q}B_q}{\kappa'}\right)^q\left(2(8(q-1))^{q-1} + 32\right)$$

$$\leq 128\left(\frac{\eta^{1-1/q}B_q}{\kappa'}\right)^q,$$

which is the desired result. $\qquad\square$

## D.6 Proof of Proposition 2

We use the invariance of $\pi_{\beta,p}$ by the transition kernel $P_{\beta,p}$. For real $\lambda$, this implies the equality

$$\mathbb{E}\exp\left(\lambda^2\|\theta - \theta^\star\|^2\right) = \mathbb{E}\exp\left(\lambda^2\|\theta - \alpha_\theta\beta G(\theta) - \theta^\star\|^2\right). \tag{39}$$

We then write:

$$\left\|\theta - \alpha_\theta\beta G(\theta) - \theta^\star\right\|^2 = \left\|\theta - \overline{\alpha}_\theta\beta\nabla\mathcal{L}(\theta) - \theta^\star\right\|^2 + \beta^2\left\|\alpha_\theta G(\theta) - \overline{\alpha}_\theta\nabla\mathcal{L}(\theta)\right\|^2$$
$$-2\beta\left\langle\theta - \overline{\alpha}_\theta\beta\nabla\mathcal{L}(\theta) - \theta^\star, \alpha_\theta G(\theta) - \mathbb{E}[\alpha_\theta G(\theta)] + \mathbb{E}[\alpha_\theta G(\theta)] - \overline{\alpha}_\theta\nabla\mathcal{L}(\theta)\right\rangle.$$

We also have the inequality

$$\left\|\alpha_\theta G(\theta) - \overline{\alpha}_\theta\nabla\mathcal{L}(\theta)\right\|^2 \leq 2\left\|\alpha_\theta G(\theta) - \mathbb{E}[\alpha_\theta G(\theta)]\right\|^2 + 2\left\|\mathbb{E}[\alpha_\theta G(\theta)] - \overline{\alpha}_\theta\nabla\mathcal{L}(\theta)\right\|^2.$$

Conditioning upon $\theta$, we have that $\|\alpha_\theta G(\theta) - \mathbb{E}[\alpha_\theta G(\theta)]\| \leq 2\tau_\theta \leq 2\overline{\tau}$. Moreover, the vector $\alpha_\theta G(\theta) - \mathbb{E}[\alpha_\theta G(\theta)]$ is centered, therefore it is sub-Gaussian with constant $2\overline{\tau}$ (see (Vershynin, 2018, Proposition 2.5.2)). Still conditioning on $\theta$, using these two properties and a Cauchy-Schwarz inequality, we find:

$$\mathbb{E}\exp\left(\lambda^2\left(2\beta^2\left\|\alpha_\theta G(\theta) - \mathbb{E}[\alpha_\theta G(\theta)]\right\|^2 - 2\beta\left\langle\theta - \overline{\alpha}_\theta\beta\nabla\mathcal{L}(\theta) - \theta^\star, \alpha_\theta G(\theta) - \mathbb{E}[\alpha_\theta G(\theta)]\right\rangle\right)\right)$$
$$\leq \exp\left(8\lambda^2\beta^2\overline{\tau}^2 + 16\lambda^4\beta^2\overline{\tau}^2\left\|\theta - \overline{\alpha}_\theta\beta\nabla\mathcal{L}(\theta) - \theta^\star\right\|^2\right).$$

Putting everything together in (39), we get:

$$\mathbb{E}\exp(\lambda^2\|\theta - \theta^\star\|^2) \leq \mathbb{E}\exp\left(\lambda^2\left((1 + 16\lambda^2\beta^2\overline{\tau}^2)\left\|\theta - \overline{\alpha}_\theta\beta\nabla\mathcal{L}(\theta) - \theta^\star\right\|^2 + 8\beta^2\overline{\tau}^2\right.\right.$$
$$\left.\left. + 2\beta^2\left\|\mathbb{E}[\alpha_\theta G(\theta)] - \overline{\alpha}_\theta\nabla\mathcal{L}(\theta)\right\|^2 + 2\beta\left\|\theta - \overline{\alpha}_\theta\beta\nabla\mathcal{L}(\theta) - \theta^\star\right\|\left\|\mathbb{E}[\alpha_\theta G(\theta)] - \overline{\alpha}_\theta\nabla\mathcal{L}(\theta)\right\|\right)\right)$$
$$\leq \mathbb{E}\exp\left(\lambda^2\left((1 + 16\lambda^2\beta^2\overline{\tau}^2 + \epsilon)\left\|\theta - \overline{\alpha}_\theta\beta\nabla\mathcal{L}(\theta) - \theta^\star\right\|^2 + 8\beta^2\overline{\tau}^2\right.\right.$$
$$\left.\left. + 2\beta^2(1 + 1/(2\epsilon))\left\|\mathbb{E}[\alpha_\theta G(\theta)] - \overline{\alpha}_\theta\nabla\mathcal{L}(\theta)\right\|^2\right)\right),$$

where we used Fact 1. Now, recalling that $\overline{\alpha}_\theta \geq p$, we set $\epsilon = \overline{\alpha}_\theta \beta \mu / 2$ and restrict $\lambda$ to $\lambda \leq (4\overline{\tau}\sqrt{2\beta/p\mu})^{-1}$ to find:

$$
\begin{aligned}
\mathbb{E} \exp \left(\lambda^2 \|\theta - \theta^\star\|^2\right) &\leq \mathbb{E} \exp \left(\lambda^2 \big((1 + 16\lambda^2\beta^2\overline{\tau}^2 + \epsilon)\|\theta - \overline{\alpha}_\theta\beta\nabla\mathcal{L}(\theta) - \theta^\star\|^2 + 8\beta^2\overline{\tau}^2 \right. \\
&\quad \left. + 2\beta^2(1 + 1/(2\epsilon))\|\mathbb{E}[\alpha_\theta G(\theta)] - \overline{\alpha}_\theta\nabla\mathcal{L}(\theta)\|^2\big)\right) \\
&\leq \mathbb{E} \exp \left(\lambda^2 \Big((1+\overline{\alpha}_\theta\beta\mu)\big\|\theta-\overline{\alpha}_\theta\beta\nabla\mathcal{L}(\theta)-\theta^\star\big\|^2 + 8\beta^2\overline{\tau}^2 + \frac{4\beta}{\overline{\alpha}_\theta\mu}\big\|\mathbb{E}[\alpha_\theta G(\theta)] - \overline{\alpha}\nabla\mathcal{L}(\theta)\big\|^2\Big)\right) \\
&\leq \mathbb{E} \exp \left(\lambda^2 \Big((1 + \overline{\alpha}_\theta\beta\mu)(1 - \overline{\alpha}_\theta\beta\mu)^2\|\theta - \theta^\star\|^2 + 8\beta^2\overline{\tau}^2 + \frac{4\beta}{\overline{\alpha}_\theta\mu}\big((1 - p)^{1-1/q}\overline{B}_q\big)^2\Big)\right) \\
&\leq \mathbb{E} \exp \left(\lambda^2 \big((1 - \overline{\alpha}_\theta\beta\mu)(1 - (\overline{\alpha}_\theta\beta\mu)^2)\|\theta - \theta^\star\|^2 + 4\beta^2(2\overline{\tau}^2 + \overline{B}_q^2/p)\big)\right),
\end{aligned}
$$

where we used Lemma 1, inequality (21) from Lemma 2 (recall that $\eta = 0$ in this context) and the imposed bound relating $\beta$ and $p$.

Finally, using Jensen's inequality and the fact that $\overline{\alpha}_\theta \geq p \geq 1/2$ we get:

$$
\mathbb{E} \exp \left(\lambda^2 \|\theta - \theta^\star\|^2\right) \leq \exp \left(\frac{8\lambda^2\beta^2\big(\overline{\tau}^2 + \overline{B}_q^2\big)}{\overline{\alpha}\beta\mu + (\overline{\alpha}\beta\mu)^2 - (\overline{\alpha}\beta\mu)^3}\right) \leq \exp \left(\frac{8\lambda^2\beta\big(\overline{\tau}^2 + \overline{B}_q^2\big)}{p\mu}\right),
$$

which concludes the first part of the proof. We now consider the corrupted case $\eta > 0$. Let $\lambda > 0$ and write :

$$
\begin{aligned}
\mathbb{E}\big[\exp\big(\lambda\|\theta - \theta^\star\|\big)\big] &= \mathbb{E}\big[\exp\big(\lambda\|\theta - \alpha_\theta\beta G(\theta) - \theta^\star\|\big)\big] \\
&\leq \eta\mathbb{E}\big[\exp\big(\lambda\|\theta-\alpha_\theta\beta\check{G}(\theta)-\theta^\star\|\big)\big] + (1-\eta)\mathbb{E}\big[\exp\big(\lambda\|\theta-\alpha_\theta\beta\widetilde{G}(\theta)-\theta^\star\|\big)\big] \\
&\leq \eta\mathbb{E}\big[\exp\big(\lambda(\|\theta - \theta^\star\| + \beta\tau_\theta)\big)\big] + (1-\eta)\mathbb{E}\big[\exp\big(\lambda\|\theta - \overline{\alpha}_\theta\beta\nabla\mathcal{L}(\theta) - \theta^\star\| \\
&\quad + \lambda\beta\big(\big\|\alpha_\theta\widetilde{G}(\theta) - \mathbb{E}[\alpha_\theta\widetilde{G}(\theta)]\big\| + \big\|\mathbb{E}[\alpha_\theta\widetilde{G}(\theta)] - \overline{\alpha}_\theta\nabla\mathcal{L}(\theta)\big\|\big)\big)\big] \\
&\leq \eta\mathbb{E}\big[\exp\big(\lambda\|\theta - \theta^\star\|\big)\big]e^{\lambda\beta\overline{\tau}} \\
&\quad + (1-\eta)\mathbb{E}\big[\exp\big(\lambda(1 - p\beta\mu)\|\theta - \theta^\star\|\big)\big] \exp\big(\lambda\beta(2\overline{\tau} + (1-p)^{1-1/q}\overline{B}_q)\big),
\end{aligned}
$$

where we used Lemma 1, the inequality $\alpha_\theta \geq p$, the inequality $\big\|\alpha_\theta\widetilde{G}(\theta) - \mathbb{E}[\alpha_\theta\widetilde{G}(\theta)]\big\| \leq 2\tau_\theta \leq 2\overline{\tau}$ and (21) from Lemma 2. Using Hölder's inequality, this leads to :

$$
\mathbb{E}\big[\exp\big(\lambda\|\theta - \theta^\star\|\big)\big] \leq \left(\frac{1 - \eta}{1 - \eta e^{\lambda\beta\overline{\tau}}}\right)^{1/(p\beta\mu)} \exp\big(\lambda(2\overline{\tau} + (1-p)^{1-1/q}\overline{B}_q)/(p\mu)\big)
$$

Now we use the inequality $\log\big(\frac{1-\eta}{1-\eta e^{\lambda\beta\overline{\tau}}}\big) \leq \frac{\beta\overline{\tau}\lambda/\log(1/\eta)^2}{1-\beta\overline{\tau}\lambda/\log(1/\eta)}$ valid for $\lambda \geq 0$ which leads to:

$$
\left(\frac{1 - \eta}{1 - \eta e^{\lambda\beta\overline{\tau}}}\right)^{1/(p\beta\mu)} \leq \exp\left(\frac{2\lambda\overline{\tau}}{p\mu\log(1/\eta)^2}\right) \quad \text{for} \quad 0 \leq \lambda \leq \frac{\log(1/\eta)}{2\beta\overline{\tau}}.
$$

Using that $\eta < 1/2$, we find that for $0 \leq \lambda \leq \frac{\log(1/\eta)}{2\beta\overline{\tau}}$, the following inequality holds :

$$
\begin{aligned}
\mathbb{E}[\exp(\lambda\|\theta - \theta^\star\|)] &\leq \exp\left(\frac{\lambda}{p\mu}\Big(2\overline{\tau}\Big(1 + \frac{1}{\log(1/\eta)^2}\Big) + (1-p)^{1-1/q}\overline{B}_q\Big)\right) \\
&\leq \exp\left(\frac{\lambda}{p\mu}\big(7\overline{\tau} + (1-p)^{1-1/q}\overline{B}_q\big)\right).
\end{aligned}
$$

Noticing that $\beta \leq \frac{1}{\mu}$ allows to finish the proof.

## D.7 Unimprovability of the sub-exponential property for $\eta > 0$

We consider the Markov chain

$$
X_{t+1} = \begin{cases} \alpha X_t + \xi & \text{w.p.} \quad 1 - \eta \\ X_t + \tau & \text{w.p.} \quad \eta \end{cases}
$$

Assuming that the distribution of the noise $\xi$ has a density, one can show that the chain is aperiodic and satisfies a minorization property as in the proof of Theorem 1 (see Appendix D.3).

Defining $V(x) = 1 + x$, we can show that $V$ verifies a geometric drift property similar to (25). Consequently, Theorem 15.0.1 of (Meyn & Tweedie, 1993) applies to the chain $(X_t)_{t \geq 0}$ and implies that it converges geometrically to a limit distribution $\pi$ analogously to the claim of Theorem 1.

We denote $M_k^k = \mathbb{E}|X|^k$ the absolute moments of $X$ for $k \geq 1$ and show that $M_k = \Omega(k)$ (we merely provide a sketch and do not attempt to explicitly compute the involved constants). For $X \sim \pi$ following the invariant measure, using the recursion defining $X_t$ and the positivity of $\xi$, it is easy to establish the inequality for $k \geq 1$

$$(1 - \eta)(1 - \alpha^k) M_k^k \geq \eta \sum_{j=1}^k \binom{k}{j} \tau^j M_{k-j}^{k-j}$$

where one may use the convention that $M_0 = 1$. We now postulate the induction hypothesis $M_j \geq Cj$ up to $j = k - 1$ for some $k > 1$ and $C > 0$. Using Stirling's formula, we find:

$$\frac{(1 - \eta)(1 - \alpha^k)}{\eta} M_k^k \geq \sum_{j=1}^k \binom{k}{j} \tau^j \left( C(k - j) \right)^{k-j} = \sum_{j=1}^k \frac{k!}{j!(k-j)!} \tau^j \left( C(k - j) \right)^{k-j}$$

$$\gtrsim \sum_{j=1}^k \sqrt{\frac{k}{j(k-j)}} \frac{k^k \tau^j \left( C(k - j) \right)^{k-j}}{j^j (k-j)^{k-j}} = \sum_{j=1}^k \sqrt{\frac{k}{j(k-j)}} \left( \frac{\tau}{j} \right)^j k^k C^{k-j}$$

$$\geq \frac{\tau}{C} (Ck)^k$$

where $\gtrsim$ denotes an inequality up to a universal constant and we took the term $j = 1$ in the last step. It is only left to set $C$ small enough such that $\frac{\tau \eta}{C(1-\eta)(1-\alpha)} \geq 1$ in order to finish the induction. It follows that $M_k = \Omega(k)$ implying that $\pi$ may be sub-exponential but cannot be sub-Gaussian since that would require $M_k = \mathcal{O}(\sqrt{k})$ (see (Vershynin, 2018, Chapter 2) for a reference).

### D.8  Proof of Corollary 1

We need the following lemma.

**Lemma 4.** *Let $X$ be a real sub-Gaussian random variable with constant $K$ then, with probability at least $\delta$, we have :*

$$|X| \leq K \sqrt{\log(e/\delta)}$$

*Proof.* Using Chernoff's method, we find for $t > 0$ and $\lambda > 0$:

$$\mathbb{P}\left( |X| > t \right) = \mathbb{P}\left( \lambda^2 X^2 > \lambda^2 t^2 \right) = \mathbb{P}\left( \exp(\lambda^2 X^2) > \exp(\lambda^2 t^2) \right)$$

$$\leq \mathbb{E} \exp(\lambda^2 X^2) e^{-\lambda^2 t^2} \leq \exp\left( \lambda^2 (K^2 - t^2) \right).$$

Choosing $\lambda = 1/K$, we have $\exp\left( 1 - (t/K)^2 \right) \leq \delta \iff t \geq K\sqrt{\log(e/\delta)}$ and the result follows. $\qquad\square$

By Theorem 1, the Markov chain $(\theta_t)_{t \geq 0}$ is geometrically converging to the invariant distribution $\pi_{\beta,p}$ w.r.t. the Total Variation distance so that for any event $\mathcal{E} \in \mathcal{B}(\mathbb{R}^d)$, we have:

$$\left| \mathbb{P}\left( \theta_T \in \mathcal{E} \right) - \mathbb{P}_{\theta \sim \pi_{\beta,p}}\left( \theta \in \mathcal{E} \right) \right| \leq M \rho^T V(\theta_0). \tag{40}$$

Proposition 2 states that, in the absence of corruption, for $\theta \sim \pi_{\beta,p}$, the variable $\|\theta - \theta^\star\|$ is sub-Gaussian with constant $K = 4\sqrt{\frac{2\beta(B_g^2 + \overline{\tau}^2)}{p\mu}}$. It is only left to combine this conclusion with Lemma 4 in order to obtain the claimed bound.

### D.9 Proof of Corollary 2

We assume without loss of generality that $T$ is a multiple of $N$. Note that according to the assumptions, the estimators $\theta_T^{(n)}$ for $n \in [\![N]\!]$ are independent and for each $n$. For positive $\epsilon < 1$ define the events $E_n := \left\{ \|\theta_T^{(n)} - \theta^\star\| \leq \frac{\eta^{1-\frac{1}{q}} B_q \sqrt{20}}{\kappa\epsilon} \right\}$. We first assume that $\sum_{n=1}^N \mathbf{1}_{E_n} > N/2$ then there exists $i' \in [\![N]\!]$ such that

$$r_{N/2}^{\widehat{(i)}} = \left\|\theta_T^{\widehat{(i)}} - \theta_T^{(i')}\right\| \leq \left\|\theta_T^{\widehat{(i)}} - \theta^\star\right\| + \left\|\theta_T^{(i')} - \theta^\star\right\| \leq 2\frac{\eta^{1-\frac{1}{q}} B_q \sqrt{20}}{\kappa\epsilon}.$$

Moreover, among the $N/2$ estimators closest to $\theta_T^{\widehat{(i)}}$, at least one of them $\theta_T^{(i'')}$ satisfies $\|\theta_T^{(i'')} - \theta^\star\| \leq \frac{\eta^{1-\frac{1}{q}} B_q \sqrt{20}}{\kappa\epsilon}$ thus we find :

$$\begin{aligned}
\left\|\widehat{\theta} - \theta^\star\right\| = \left\|\theta_T^{\widehat{(i)}} - \theta^\star\right\| &\leq \left\|\theta_T^{\widehat{(i)}} - \theta_T^{(i'')}\right\| + \left\|\theta_T^{(i'')} - \theta^\star\right\| \\
&\leq r_{N/2}^{\widehat{(i)}} + \frac{\eta^{1-\frac{1}{q}} B_q \sqrt{20}}{\kappa\epsilon} \leq 3\frac{\eta^{1-\frac{1}{q}} B_q \sqrt{20}}{\kappa\epsilon}.
\end{aligned} \tag{41}$$

Notice that setting $\epsilon = 1/2$ immediately yields (13). We now show that $\sum_{n=1}^N \mathbf{1}_{E_n} > N/2$ happens with high probability. Thanks to Theorem 1 and Proposition 1, we have:

$$\mathbb{P}(\overline{E}_n) \leq \epsilon^2 + \underbrace{M\rho^{T/N}\left(1 + \|\theta_0 - \theta^\star\|^2\right)}_{\epsilon'}.$$

Consequently, the variables $\mathbf{1}_{\overline{E}_n}$ are stochastically dominated by Bernoulli variables with parameter $\epsilon^2 + \epsilon'$ so that their sum is stochastically dominated by a Binomial random variable $S := \mathrm{Bin}(N, \epsilon^2 + \epsilon')$. We compute :

$$\begin{aligned}
\mathbb{P}\left(\sum_{n=1}^N \mathbf{1}_{E_n} < N/2\right) = \mathbb{P}\left(\sum_{n=1}^N \mathbf{1}_{\overline{E}_n} > N/2\right) &\leq \mathbb{P}\left(S - \mathbb{E}S > N/2 - (\epsilon^2 + \epsilon')N\right) \\
&\leq \exp\left(-2N(1/2 - \epsilon^2 - \epsilon')^2\right) \\
&\leq \exp\left(-2N(1/2 - 1/4 - M\rho^{T/N}(1 + \|\theta_0 - \theta^\star\|^2))^2\right) \\
&\leq \exp\left(-2N(1/4 - 1/15)^2\right) \leq \exp\left(-\log(1/\delta)\right) = \delta
\end{aligned}$$

where we used Hoeffding's inequality, the choice $\epsilon = 1/2$ and the fact that our condition on $T$ implies $M\rho^{T/N}(1 + \|\theta_0 - \theta^\star\|^2))^2 \leq 1/15$. The last inequalities result from our condition on $N$.

### D.10 Proof of Theorem 2

As previously done in the proof of Theorem 1, we show that the Markov chain is aperiodic. Note that since $\mathcal{L}$ has finite lower bound $\inf_\theta \mathcal{L}$, we can replace it with $1 + \mathcal{L}(\theta) - \inf_\theta \mathcal{L}$ and assume it is positive in the rest of the proof without loss of generality. We will now show that it satisfies a drift property. Let $\theta \in \Theta$ be fixed, using Assumptions 1 and 3, we have :

$$\begin{aligned}
\mathbb{E}\left[\mathcal{L}\left(\theta - \alpha_\theta \beta G(\theta)\right)\right] - \mathcal{L}(\theta) &\leq \mathbb{E}\left[-\beta\langle\nabla\mathcal{L}(\theta), \alpha_\theta G(\theta)\rangle + \frac{L\beta^2}{2}\|\alpha_\theta G(\theta)\|^2\right] \\
&\leq \mathbb{E}\left[\eta\left(-\beta\langle\nabla\mathcal{L}(\theta), \alpha_\theta \breve{G}(\theta)\rangle + \frac{L\beta^2}{2}\|\alpha_\theta \breve{G}(\theta)\|^2\right) + \right. \\
&\qquad \left. (1-\eta)\left(-\beta\langle\nabla\mathcal{L}(\theta), \alpha_\theta \widetilde{G}(\theta)\rangle + \frac{L\beta^2}{2}\|\alpha_\theta \widetilde{G}(\theta)\|^2\right)\right]
\end{aligned}$$

Note that we have $\|\alpha_\theta \widetilde{G}(\theta)\|, \|\alpha_\theta \check{G}(\theta)\| \le \tau_\theta$, therefore

$$\mathbb{E}\big[\mathcal{L}\big(\theta - \alpha_\theta \beta G(\theta)\big)\big] - \mathcal{L}(\theta) \le \mathbb{E}\Big[\eta\Big(-\beta\langle\nabla\mathcal{L}(\theta), \alpha_\theta\check{G}(\theta)\rangle\Big)+$$
$$(1-\eta)\Big(-\beta\langle\nabla\mathcal{L}(\theta), \alpha_\theta\widetilde{G}(\theta)\rangle\Big)\Big] + \frac{L\beta^2\tau_\theta^2}{2}.$$

Further, we write

$$\mathbb{E}\big[\langle\nabla\mathcal{L}(\theta), \alpha_\theta\widetilde{G}(\theta)\rangle\big] = \langle\nabla\mathcal{L}(\theta), \mathbb{E}[\alpha_\theta\widetilde{G}(\theta)] - \overline{\alpha}_\theta\nabla\mathcal{L}(\theta) + \overline{\alpha}_\theta\nabla\mathcal{L}(\theta)\rangle$$
$$= \overline{\alpha}_\theta\big\|\nabla\mathcal{L}(\theta)\big\|^2 + \langle\nabla\mathcal{L}(\theta), \mathbb{E}[\alpha_\theta\widetilde{G}(\theta)] - \overline{\alpha}_\theta\nabla\mathcal{L}(\theta)\rangle.$$

Plugging back and using Cauchy-Schwartz and the inequality $\|\alpha_\theta\check{G}(\theta)\| \le \tau_\theta$ leads to

$$\mathbb{E}\big[\mathcal{L}\big(\theta - \alpha_\theta\beta G(\theta)\big)\big] - \mathcal{L}(\theta) \le -\eta\beta\langle\nabla\mathcal{L}(\theta), \mathbb{E}[\alpha_\theta\check{G}(\theta)]\rangle - \beta(1-\eta)\overline{\alpha}_\theta\big\|\nabla\mathcal{L}(\theta)\big\|^2-$$
$$\beta(1-\eta)\langle\nabla\mathcal{L}(\theta), \mathbb{E}[\alpha_\theta\widetilde{G}(\theta)] - \overline{\alpha}_\theta\nabla\mathcal{L}(\theta)\rangle + \frac{L\beta^2\tau_\theta^2}{2}$$
$$\le \eta\beta\tau_\theta\|\nabla\mathcal{L}(\theta)\| - \beta(1-\eta)\overline{\alpha}_\theta\|\nabla\mathcal{L}(\theta)\|^2+$$
$$\beta(1-\eta)\|\nabla\mathcal{L}(\theta)\|\big\|\mathbb{E}[\alpha_\theta\widetilde{G}(\theta)] - \overline{\alpha}_\theta\nabla\mathcal{L}(\theta)\big\| + \frac{L\beta^2\tau_\theta^2}{2}.$$

We now use the inequalities $\tau_\theta \le \|\nabla\mathcal{L}(\theta)\| + Q_p(\|\varepsilon_\theta\|)$ (see Lemma 2) and $(a+b)^2 \le 2a^2 + 2b^2$, to find that

$$\mathbb{E}\big[\mathcal{L}\big(\theta - \alpha_\theta\beta G(\theta)\big)\big] - \mathcal{L}(\theta) \le -\beta\|\nabla\mathcal{L}(\theta)\|^2\big((1-\eta)\overline{\alpha}_\theta - L\beta - \eta\big) + L\beta^2 Q_p(\|\varepsilon_\theta\|)^2+$$
$$\beta\|\nabla\mathcal{L}(\theta)\|\big(\eta Q_p(\|\varepsilon_\theta\|) + (1-\eta)\big\|\mathbb{E}[\alpha_\theta\widetilde{G}(\theta)] - \overline{\alpha}_\theta\nabla\mathcal{L}(\theta)\big\|\big).$$

Next, letting $\epsilon > 0$, and using Fact 1 gives

$$\|\nabla\mathcal{L}(\theta)\|\big(\eta Q_p(\|\varepsilon_\theta\|) + (1-\eta)\big\|\mathbb{E}[\alpha_\theta\widetilde{G}(\theta)] - \overline{\alpha}_\theta\nabla\mathcal{L}(\theta)\big\|\big) \le \epsilon\|\nabla\mathcal{L}(\theta)\|^2/2+$$
$$\big(\eta Q_p(\|\varepsilon_\theta\|) + (1-\eta)\big\|\mathbb{E}[\alpha_\theta\widetilde{G}(\theta)] - \overline{\alpha}_\theta\nabla\mathcal{L}(\theta)\big\|\big)^2/(2\epsilon).$$

We now plug back with the choice $\epsilon = p(1-\eta)/2$ and use that $\overline{\alpha}_\theta \ge p$ to find

$$\mathbb{E}\big[\mathcal{L}\big(\theta - \alpha_\theta\beta G(\theta)\big)\big] - \mathcal{L}(\theta) \le -\beta\|\nabla\mathcal{L}(\theta)\|^2\big(3p(1-\eta)/4 - L\beta - \eta\big) + L\beta^2 Q_p(\|\varepsilon_\theta\|)^2+$$
$$\frac{\beta\big(\eta Q_p(\|\varepsilon_\theta\|) + (1-\eta)\big\|\mathbb{E}[\alpha_\theta\widetilde{G}(\theta)] - \overline{\alpha}_\theta\nabla\mathcal{L}(\theta)\big\|\big)^2}{p(1-\eta)}.$$

Finally, we use Lemma 2 to bound the terms $Q_p(\|\varepsilon_\theta\|)$ and $\big\|\mathbb{E}[\alpha_\theta\widetilde{G}(\theta)] - \overline{\alpha}_\theta\nabla\mathcal{L}(\theta)\big\|$ leading to

$$\mathbb{E}\big[\mathcal{L}\big(\theta - \alpha_\theta\beta G(\theta)\big)\big] - \mathcal{L}(\theta) \le -\beta\|\nabla\mathcal{L}(\theta)\|^2\big(3p(1-\eta)/4 - L\beta - \eta\big) + L\beta^2(1-p)^{-\frac{2}{q}}B_q^2+$$
$$\frac{\beta B_q^2\big(\eta(1-p)^{-\frac{1}{q}} + (1-\eta)\eta^{1-\frac{1}{q}}\big)^2}{p(1-\eta)}$$
$$\le -\beta\|\nabla\mathcal{L}(\theta)\|^2\big(3p(1-\eta)/4 - L\beta - \eta\big)+$$
$$\frac{\beta B_q^2\big((1-p)^{-\frac{2}{q}}(L\beta + 2\eta^2) + 2\eta^{2-\frac{2}{q}}\big)}{p(1-\eta)}. \tag{42}$$

By assumption, we have $3p(1-\eta)/4 - L\beta - \eta > 0$. Define the quantity $\xi = \frac{B_q^2\big((1-p)^{-\frac{2}{q}}(L\beta + 2\eta^2) + 2\eta^{2-\frac{2}{q}}\big)}{p(1-\eta)}$ and the set $\mathcal{C} := \Big\{\theta \in \mathbb{R}^d : \frac{1}{2}\|\nabla\mathcal{L}(\theta)\|^2 \le \frac{\xi}{3p(1-\eta)/4 - L\beta - \eta}\Big\}$. By assumption, $\mathcal{C}$ is bounded and it is clear that the right hand side in (42) is negative outside $\mathcal{C}$. Define the function $V(\theta) = \mathcal{L}(\theta)/(\beta\xi)$, which is positive and satisfies:

$$\Delta V(\theta) \le -1 + 2 \cdot \mathbf{1}_{\theta\in\mathcal{C}}. \tag{43}$$

In addition, we show similarly to Theorem 1 that the set $\mathcal{C}$ is *small* and, therefore, also *petite* according to the definitions of (Meyn & Tweedie, 1993, Chapter 5). Since $V$ is everywhere finite and bounded on $\mathcal{C}$ (because the latter is compact), the conditions of (Meyn & Tweedie, 1993, Theorem 11.3.4) are fulfilled implying that the chain is Harris recurrent.

We have shown that the Markov chain verifies the fourth condition of (Meyn & Tweedie, 1993, Theorem 13.0.1). This allows us to conclude that the Markov chain is ergodic i.e. we have for any initial measure $\lambda$ that $\|\lambda P^t - \pi_{\beta,p}\|_{\mathrm{TV}} \to 0$ and the following sum is finite

$$\sum_t \|\lambda P_{\beta,p}^t - \pi_{\beta,p}\|_{\mathrm{TV}} < \infty.$$

In addition, by (Meyn & Tweedie, 1993, Proposition 13.3.2) the terms in the above sum are non-increasing which implies that $\|\lambda P_{\beta,p}^t - \pi_{\beta,p}\|_{\mathrm{TV}} = \mathcal{O}(t^{-1})$ and the result follows.

### D.11 Proof of Proposition 3

By Theorem 2, the assumptions imply that the Markov chain $(\theta_t)_{t\geq 0}$ converges to an invariant distribution $\pi_{\beta,p}$. For $\theta \sim \pi_{\beta,1-\eta}$, by invariance of $\pi_{\beta,1-\eta}$, we have that the variables $\mathcal{L}(\theta - \alpha_\theta \beta G(\theta))$ and $\mathcal{L}(\theta)$ are identically distributed. Taking the expectation w.r.t. $\theta$, this implies the identity $\mathbb{E}[\mathcal{L}(\theta - \alpha_\theta \beta G(\theta))] = \mathbb{E}[\mathcal{L}(\theta)]$. Plugging into Inequality (42) from the proof of Theorem 2, we find

$$
\begin{aligned}
\mathbb{E}\left[\|\nabla \mathcal{L}(\theta)\|^2\right] &\leq \frac{B_q^2\left((1-p)^{-\frac{2}{q}}(L\beta + 2\eta^2) + 2\eta^{2-\frac{2}{q}}\right)}{p(1-\eta)\left(3p(1-\eta)/4 - L\beta - \eta\right)} \\
&\leq \frac{B_q^2\left(3(1-p)^{-\frac{2}{q}}\eta^2 + 2\eta^{2-\frac{2}{q}}\right)}{p(1-\eta)\left(3p(1-\eta)/4 - L\beta - \eta\right)} \\
&\leq \frac{5B_q^2\eta^{2-\frac{2}{q}}}{p(1-\eta)\left(3p(1-\eta)/4 - L\beta - \eta\right)}
\end{aligned}
$$

where we used the choices $\beta \leq \frac{\eta^2}{L}$ and $p = 1 - \eta$.

