# Robust Stochastic Optimization via Gradient Quantile Clipping

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

**Broader Impact Statement**

In this optional section, TMLR encourages authors to discuss possible repercussions of their work, notably any potential negative impact that a user of this research should be aware of. Authors should consult the TMLR Ethics Guidelines available on the TMLR website for guidance on how to approach this subject.

**Author Contributions**

If you'd like to, you may include a section for author contributions as is done in many journals. This is optional and at the discretion of the authors. Only add this information once your submission is accepted and deanonymized.

**Acknowledgments**

Use unnumbered third level headings for the acknowledgments. All acknowledgments, including those to funding agencies, go at the end of the paper. Only add this information once your submission is accepted and deanonymized.

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

# Supplementary Material
## Robust Stochastic Optimization via Gradient Quantile Clipping

## A  Additional experimental results

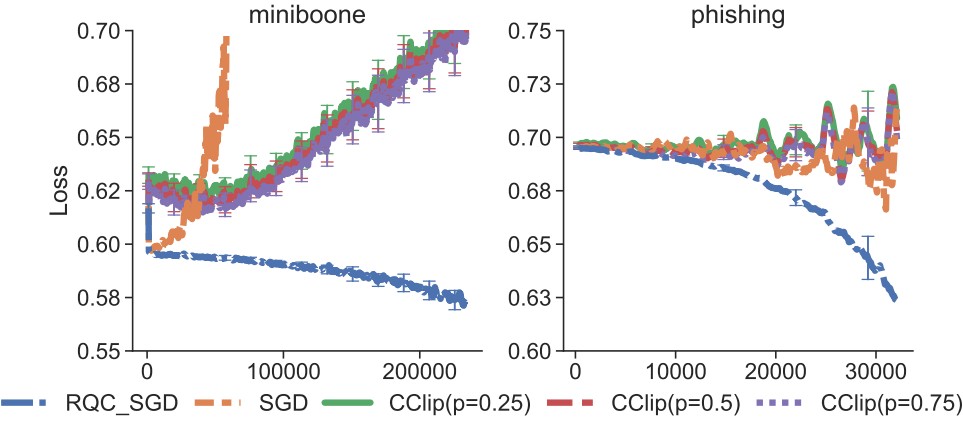

Figure 3: Evolution of the test loss ($y$-axis) against iteration $t$ ($x$-axis) for the training of a single hidden layer network on additional real world classification datasets (average over 20 runs).

**Classification with shallow networks.** We performed the same experiment using two additional datasets. The results are displayed on Figure 3 and corroborate our statements in the main paper.

**Expectation estimation.** We estimate the expectation of a random vector $X$ by minimizing the objective $\mathcal{L}(\theta) = \frac{1}{2}\|\theta - \theta^\star\|^2$ with $\theta^\star = \mathbb{E}[X]$ using a stream of both corrupted and heavy-tailed samples, see Appendix B for details. We run RQC-SGD (Algorithm 2) and compare it to an online version of geometric and coordinate-wise Median-Of-Means (GMOM and CMOM (Cardot et al., 2017; 2013)) which use block sample means to minimize an $L_1$ objective (see Appendix B). Although these estimators are a priori not robust to $\eta$-corruption, we ensure that their estimates are meaningful by limiting $\eta$ to 4% and using blocks of 10 samples. Thus, blocks are corrupted with probability $< 1/2$ so that the majority contains only true samples. Figure 4 displays the evolution of $\|\theta_t - \theta^\star\|$ for each method averaged over 100 runs for increasing $\eta$ and constant step size. We also display a single run for $\eta = 0.04$. We observe that RQC-SGD is only weakly affected by the increasing corruption whereas the performance of GMOM and CMOM quickly degrades with $\eta$, leading to unstable estimates.

## B  Experimental details

As previously mentioned, the dimension is set to $d = 128$ in our experiments with synthetic data. We also set $\sigma_{\min} = 1$ and $\sigma_{\max} = 5$ as minimum and maximum scaling factors. For all tasks and algorithms, the optimization starts from $\theta_0 = 0$.

**Bookkeeping in RQC-SGD** The buffer in Algorithm 2 stores values in sorted order along with their "ages". The most recent and oldest values have ages 0 and $S - 1$ respectively. At each iteration, a new gradient is received, all ages are incremented and the oldest value is replaced by the new one with age 0. The latter is then sorted using one iteration of insertion sort. The estimate $\widehat{Q}_p$ is retrieved at each iteration as the value at position $\lfloor pS \rfloor$.

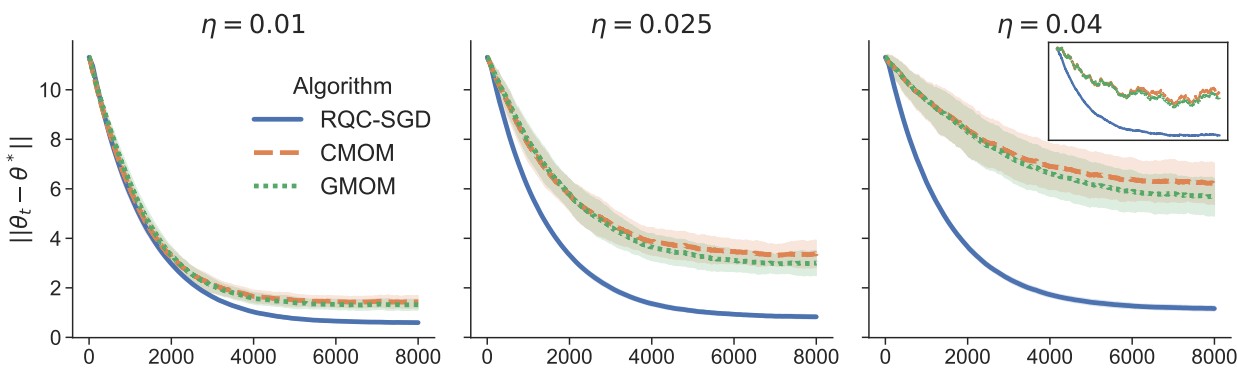

Figure 4: Evolution of $\|\theta_t - \theta^\star\|$ ($y$-axis) against iteration $t$ ($x$-axis) for the expectation estimation task, averaged over 100 runs at different corruption levels $\eta$ (bands widths correspond to the standard deviation of the 100 runs). For $\eta = 0.04$, the evolution on a single run is also displayed. We observe good performance for RQC-SGD for increasing $\eta$ while CMOM and GMOM are more sensitive.

### B.1 Mean estimation

**Data generation** We compute a matrix $\Sigma = (AA^\top + A^\top A)/2$ where $A \in \mathbb{R}^{d \times d}$ is a random matrix with i.i.d centered Gaussian entries with variance $1/d$ sampled once and for all. We generate true samples as $X = \mathbf{1} + \Sigma V$ where $V$ is a vector of i.i.d symmetrized Pareto random variables with parameter 2 and $\mathbf{1} \in \mathbb{R}^d$ denotes the vector with all entries equal to 1.

We draw corrupted samples as $\check{X} = 10\check{V} - 100 \times \mathbf{1}$ where $\check{V}$ is a vector of i.i.d symmetrized Pareto variables with parameter 1.5. We use step size $\beta = 10^{-3}$.

**GMOM and CMOM** The geometric and coordinatewise Median-Of-Means estimators (GMOM and CMOM) optimize the following objectives respectively:

$$\mathbb{E}\big\|\theta - \overline{X}_{N_b}\big\|_2 \quad \text{and} \quad \mathbb{E}\big\|\theta - \overline{X}_{N_b}\big\|_1,$$

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

&\leq \eta\mathbb{E}\big[\exp\left(\lambda\|\theta - \alpha_\theta\beta\breve{G}(\theta) - \theta^\star\|\right)\big] + (1-\eta)\mathbb{E}\big[\exp\left(\lambda\|\theta - \alpha_\theta\beta\widetilde{G}(\theta) - \theta^\star\|\right)\big]\\
&\leq \eta\mathbb{E}\big[\exp\left(\lambda(\|\theta - \theta^\star\| + \beta\tau_\theta)\right)\big] + (1-\eta)\mathbb{E}\big[\exp\left(\lambda\|\theta - \overline{\alpha}_\theta\beta\nabla\mathcal{L}(\theta) - \theta^\star\|\right.\\
&\quad \left. + \lambda\beta\big(\|\alpha_\theta\widetilde{G}(\theta) - \mathbb{E}[\alpha_\theta\widetilde{G}(\theta)]\| + \|\mathbb{E}[\alpha_\theta\widetilde{G}(\theta)] - \overline{\alpha}_\theta\nabla\mathcal{L}(\theta)\|\big)\right)\big]\\
&\leq \eta\mathbb{E}\big[\exp\left(\lambda\|\theta - \theta^\star\|\right)\big]e^{\lambda\beta\overline{\tau}}\\
&\quad + (1-\eta)\mathbb{E}\big[\exp\left(\lambda(1 - p\beta\mu)\|\theta - \theta^\star\|\right)\big]\exp\left(\lambda\beta(2\overline{\tau} + (1-p)^{1-1/q}\overline{B}_q)\right),
\end{aligned}
$$

where we used Lemma 1, the inequality $\alpha_\theta \geq p$, the inequality $\|\alpha_\theta\widetilde{G}(\theta) - \mathbb{E}[\alpha_\theta\widetilde{G}(\theta)]\| \leq 2\tau_\theta \leq 2\overline{\tau}$ and (21) from Lemma 2. Using Hölder's inequality, this leads to :

$$
\mathbb{E}\big[\exp\left(\lambda\|\theta - \theta^\star\|\right)\big] \leq \left(\frac{1-\eta}{1 - \eta e^{\lambda\beta\overline{\tau}}}\right)^{1/(p\beta\mu)}\exp\left(\lambda(2\overline{\tau} + (1-p)^{1-1/q}\overline{B}_q)/(p\mu)\right)
$$

Now we use the inequality $\log\left(\frac{1-\eta}{1 - \eta e^{\lambda\beta\overline{\tau}}}\right) \leq \frac{\beta\overline{\tau}\lambda/\log(1/\eta)^2}{1 - \beta\overline{\tau}\lambda/\log(1/\eta)}$ valid for $\lambda \geq 0$ which leads to:

$$
\left(\frac{1-\eta}{1 - \eta e^{\lambda\beta\overline{\tau}}}\right)^{1/(p\beta\mu)} \leq \exp\left(\frac{2\lambda\overline{\tau}}{p\mu\log(1/\eta)^2}\right) \quad \text{for} \quad 0 \leq \lambda \leq \frac{\log(1/\eta)}{2\beta\overline{\tau}}.
$$

Using that $\eta < 1/2$, we find that for $0 \leq \lambda \leq \frac{\log(1/\eta)}{2\beta\overline{\tau}}$, the following inequality holds :

$$
\mathbb{E}[\exp(\lambda\|\theta - \theta^\star\|)] \leq \exp\left(\frac{\lambda}{p\mu}\Big(2\overline{\tau}\Big(1 + \frac{1}{\log(1/\eta)^2}\Big) + (1-p)^{1-1/q}\overline{B}_q\Big)\right)
$$

$$
\leq \exp\left(\frac{\lambda}{p\mu}\big(7\overline{\tau} + (1-p)^{1-1/q}\overline{B}_q\big)\right).
$$

Noticing that $\beta \leq \frac{1}{\mu}$ allows to finish the proof.

### D.7 Unimprovability of the sub-exponential property for $\eta > 0$

We consider the Markov chain

$$
X_{t+1} = \begin{cases} \alpha X_t + \xi & \text{w.p.} \quad 1 - \eta\\ X_t + \tau & \text{w.p.} \quad \eta \end{cases}
$$

Assuming that the distribution of the noise $\xi$ has a density, one can show that the chain is aperiodic and satisfies a minorization property as in the proof of Theorem 1 (see Appendix D.3).

Defining $V(x) = 1 + x$, we can show that $V$ verifies a geometric drift property similar to (25). Consequently, Theorem 15.0.1 of (Meyn & Tweedie, 1993) applies to the chain $(X_t)_{t \geq 0}$ and implies that it converges geometrically to a limit distribution $\pi$ analogously to the claim of Theorem 1.

We denote $M_k^k = \mathbb{E}|X|^k$ the absolute moments of $X$ for $k \geq 1$ and show that $M_k = \Omega(k)$ (we merely provide a sketch and do not attempt to explicitly compute the involved constants). For $X \sim \pi$ following the invariant measure, using the recursion defining $X_t$ and the positivity of $\xi$, it is easy to establish the inequality for $k \geq 1$

$$(1 - \eta)(1 - \alpha^k)M_k^k \geq \eta \sum_{j=1}^{k} \binom{k}{j} \tau^j M_{k-j}^{k-j}$$

where one may use the convention that $M_0 = 1$. We now postulate the induction hypothesis $M_j \geq Cj$ up to $j = k - 1$ for some $k > 1$ and $C > 0$. Using Stirling's formula, we find:

$$\frac{(1 - \eta)(1 - \alpha^k)}{\eta} M_k^k \geq \sum_{j=1}^{k} \binom{k}{j} \tau^j \big(C(k-j)\big)^{k-j} = \sum_{j=1}^{k} \frac{k!}{j!(k-j)!} \tau^j \big(C(k-j)\big)^{k-j}$$

$$\gtrsim \sum_{j=1}^{k} \sqrt{\frac{k}{j(k-j)}} \frac{k^k \tau^j \big(C(k-j)\big)^{k-j}}{j^j (k-j)^{k-j}} = \sum_{j=1}^{k} \sqrt{\frac{k}{j(k-j)}} \left(\frac{\tau}{j}\right)^j k^k C^{k-j}$$

$$\geq \frac{\tau}{C}(Ck)^k$$

where $\gtrsim$ denotes an inequality up to a universal constant and we took the term $j = 1$ in the last step. It is only left to set $C$ small enough such that $\frac{\tau \eta}{C(1-\eta)(1-\alpha)} \geq 1$ in order to finish the induction. It follows that $M_k = \Omega(k)$ implying that $\pi$ may be sub-exponential but cannot be sub-Gaussian since that would require $M_k = \mathcal{O}(\sqrt{k})$ (see (Vershynin, 2018, Chapter 2) for a reference).

### D.8   Proof of Corollary 1

We need the following lemma.

**Lemma 4.** *Let $X$ be a real sub-Gaussian random variable with constant $K$ then, with probability at least $\delta$, we have :*

$$|X| \leq K\sqrt{\log(e/\delta)}$$

*Proof.* Using Chernoff's method, we find for $t > 0$ and $\lambda > 0$:

$$\mathbb{P}\big(|X| > t\big) = \mathbb{P}\big(\lambda^2 X^2 > \lambda^2 t^2\big) = \mathbb{P}\big(\exp(\lambda^2 X^2) > \exp(\lambda^2 t^2)\big)$$

$$\leq \mathbb{E}\exp(\lambda^2 X^2)e^{-\lambda^2 t^2} \leq \exp\big(\lambda^2(K^2 - t^2)\big).$$

Choosing $\lambda = 1/K$, we have $\exp\big(1 - (t/K)^2\big) \leq \delta \iff t \geq K\sqrt{\log(e/\delta)}$ and the result follows. $\square$

By Theorem 1, the Markov chain $(\theta_t)_{t \geq 0}$ is geometrically converging to the invariant distribution $\pi_{\beta,p}$ w.r.t. the Total Variation distance so that for any event $\mathcal{E} \in \mathcal{B}(\mathbb{R}^d)$, we have:

$$\big|\mathbb{P}\big(\theta_T \in \mathcal{E}\big) - \mathbb{P}_{\theta \sim \pi_{\beta,p}}\big(\theta \in \mathcal{E}\big)\big| \leq M\rho^T V(\theta_0). \tag{40}$$

Proposition 2 states that, in the absence of corruption, for $\theta \sim \pi_{\beta,p}$, the variable $\|\theta - \theta^\star\|$ is sub-Gaussian with constant $K = 4\sqrt{\frac{2\beta(B_g^2 + \overline{\tau}^2)}{p\mu}}$. It is only left to combine this conclusion with Lemma 4 in order to obtain the claimed bound.

### D.9   Proof of Corollary 2

We assume without loss of generality that $T$ is a multiple of $N$. Note that according to the assumptions, the estimators $\theta_T^{(n)}$ for $n \in [\![N]\!]$ are independent and for each $n$. For positive $\epsilon < 1$ define the events $E_n := \big\{ \|\theta_T^{(n)} - \theta^\star\| \le \frac{\eta^{1-\frac{1}{q}} B_q \sqrt{20}}{\kappa\epsilon} \big\}$. We first assume that $\sum_{n=1}^N \mathbf{1}_{E_n} > N/2$ then there exists $i' \in [\![N]\!]$ such that

$$ r_{N/2}^{\widehat{(i)}} = \big\|\theta_T^{\widehat{(i)}} - \theta_T^{(i')}\big\| \le \big\|\theta_T^{\widehat{(i)}} - \theta^\star\big\| + \big\|\theta_T^{(i')} - \theta^\star\big\| \le 2 \frac{\eta^{1-\frac{1}{q}} B_q \sqrt{20}}{\kappa\epsilon}. $$

Moreover, among the $N/2$ estimators closest to $\theta_T^{\widehat{(i)}}$, at least one of them $\theta_T^{(i'')}$ satisfies $\|\theta_T^{(i'')} - \theta^\star\| \le \frac{\eta^{1-\frac{1}{q}} B_q \sqrt{20}}{\kappa\epsilon}$ thus we find :

$$ \big\|\widehat{\theta} - \theta^\star\big\| = \big\|\theta_T^{\widehat{(i)}} - \theta^\star\big\| \le \big\|\theta_T^{\widehat{(i)}} - \theta_T^{(i'')}\big\| + \big\|\theta_T^{(i'')} - \theta^\star\big\| $$
$$ \le r_{N/2}^{\widehat{(i)}} + \frac{\eta^{1-\frac{1}{q}} B_q \sqrt{20}}{\kappa\epsilon} \le 3 \frac{\eta^{1-\frac{1}{q}} B_q \sqrt{20}}{\kappa\epsilon}. \tag{41} $$

Notice that setting $\epsilon = 1/2$ immediately yields (13). We now show that $\sum_{n=1}^N \mathbf{1}_{E_n} > N/2$ happens with high probability. Thanks to Theorem 1 and Proposition 1, we have:

$$ \mathbb{P}(\overline{E}_n) \le \epsilon^2 + \underbrace{M\rho^{T/N}\big(1 + \|\theta_0 - \theta^\star\|^2\big)}_{\epsilon'}. $$

Consequently, the variables $\mathbf{1}_{\overline{E}_n}$ are stochastically dominated by Bernoulli variables with parameter $\epsilon^2 + \epsilon'$ so that their sum is stochastically dominated by a Binomial random variable $S := \text{Bin}(N, \epsilon^2 + \epsilon')$. We compute :