# OpenReview forum: "Robust Stochastic Optimization via Gradient Quantile Clipping"
_TMLR — Accepted by TMLR_

### Review · Reviewer_m6Va · 2024-08-05

**Summary Of Contributions:**

This paper introduces a gradient quantile clipping strategy for SGD (QC-SGD) in the setting of streaming stochastic optimization with data corruption. The contributions include:
1. Gives the result of convergence in distribution of the Markov Chain by SGD with constant step size: For both strongly convex and non-convex objective, with well chosen clipping quantile $p$, QC-SGD allows to combine both fast convergence and optimal dependence on corruption parameter $\eta$.
2. Proposes a rolling quantile procedure for the quantile estimation and confirm its robustness by numerical experiments on three stochastic optimization tasks.

**Audience:**

Yes

**Broader Impact Concerns:**

No broader impact is discussed.

**Claims And Evidence:**

Yes

**Requested Changes:**

1. Add more comparisons with [28] as above.
2. Add some details about how to choose $p$.

**Strengths And Weaknesses:**

Strengths:
1. The paper introduces gradient quantile clipping at the first time to the best of my knowledge, and provide sound theoretical guarantee for it robustness to corruptions in the data stream.
2. The paper also provide a rolling quantile procedure for simple implementation and shows its performance through numerical experiments.

Weaknesses:
1. It is mentioned in this paper several times that another work [28] (which also proposes algorithms for data corruption robust streaming estimation) is difficult to implement and compare, could the authors provide more detailed reasons about this?
2. The high probability bound given in Corollary 2 is still weaker than the results in [28] due to stronger dimension dependence, could the authors provide potential improvement on current procedure (Algorithm 1)? And could the general results be further improved for specific linear and logistic regression to achieve the similar bounds with [28]?
3. The authors' main argument that the proposed algorithm is better than [28] is that it is easy to implement without dependence on a number of unknown constants. However, it is still not clear to me how to choose $p$ in the implementation of specific tasks, which is an essential parameter for the convergence of the proposed algorithm as stated in Theorem 1 and Theorem 2.

---

> ### Author Response · Authors · 2024-08-29
> **Reply to Reviewer m6Va**
>
> We thank for time spent reading an reviewing our manuscript and for your constructive criticism.
>
> We reply below to the weaknesses you pointed out and detail the modification we brought to the paper to address them.
>
> 1. The algorithm in question is difficult to implement for a number of reasons. It requires knowledge of problem parameters like the corruption rate, desired confidence level and desired estimation precision for the computation of multiple constants. Many of these constants are vaguely required to be high enough or only specified with a $\Theta(\cdot)$ or a $\mathrm{polylog}(\cdot)$ notation and would lead to impractically high values. This would translate, for instance, into high cardinalities for the mini-batches that need to be used. The algorithm is actually an adaptation of a filtering procedure destined for an offline setting by running it on mini-batches. Beyond the fact that the filtering requires multiple passes and should not be expected to be efficient, the use of mini-batches makes the procedure rather inappropriate for streaming estimation. Moreover, the pseudocode features matrix computations which would normally imply an unacceptable $O(d^2)$ time and memory complexity. These computations need to be avoided using special methods which are presented separately and introduce further intricacies into the overall procedure. We have expanded upon the matter in the experimental section of our revision.
>
>
> 2. A possible way to improve the dimension dependence in Corollary 2 and other results in the paper is to reformulate Assumption 5 in terms of $\sup_{\|v\|=1}\mathbb{E}\big|\langle \varepsilon_{\theta}, v\rangle\big|$ and use clipping thresholds $\tau_{\theta_t}$ based on projections instead of norms. This idea is inspired from the paper "Dimension-free PAC-Bayesian bounds for the estimation of the mean of a random vector" by Catoni et al. However, this would require estimating the quantiles $\sup_{\|v\|=1}Q_p\big(|\langle \widetilde{G}(\theta_t, \zeta_t), v\rangle |\big)$ along all directions $\|v\|=1$ which is too sample hungry and computationally heavy for stochastic optimization. Moreover, this would raise a number of issues in the analysis. For instance, the upperbound in Assumption 5 would need to be redefined in a dimension-free fashion and the analysis would need to be fundamentally overhauled to avoid using Euclidean norms. The restriction to linear and logistic regression does not allow to bypass these difficulties. We have included these additional elements in the discussion following Proposition 1 in the revision.
>
>
> 3. As stated in Theorems 1 and 2, valid choices of $p$ are within the interval $[\eta, 1-\eta].$ Note however that this refers to true gradient norm quantiles $\|\widetilde{G}(\theta_t, \zeta_t)\|$ while, in practice, one only has access to $\|G(\theta_t, \zeta_t)\|$. Therefore, one should avoid values above $1-2\eta$ when an estimation of $\eta$ is available. If this is not the case, one may default to $p=1/2$ as a good initial guess then adapt based on observed performance. In our experiments, we found that small values like $p\in[0.1, 0.2]$ are best in the strongly convex case whereas, for non-strongly convex objectives, it is worth using higher values. For instance, we use $p=1-\eta-0.05$ (with known $\eta$) for logistic regression and $p=0.9$ for the single hidden layer classifier. See also the Experimental details section of our Supplementary material. We have added a paragraph in the experimental section of our revision to include more details and guidance on this matter.
>
> Note that although the ideal value for $p$ may vary depending on the problem at hand. The task of finetuning a single value in the interval $[0, 1]$ remains much simpler than handling the intricacies of the algorithm proposed in [28] for instance.

---

### Review · Reviewer_z2UW · 2024-08-09

**Summary Of Contributions:**

The paper addresses the problem of stochastic optimization under a modified heavy tails assumption on gradient estimates, with potential probabilistic corruptions of the samples. The authors propose a method based on SGD with a clipping threshold selected according to the quantiles of stochastic gradient norms. They provide several theoretical results on the convergence of the iterates' "distribution" and the deviation of the stationary measure samples from the optimum for both strongly convex and non-convex cases. Finally, a more practical algorithm variation is introduced and tested on various synthetic and small machine learning problems.

**Audience:**

Yes

**Broader Impact Concerns:**

No, for this work

**Claims And Evidence:**

Yes

**Requested Changes:**

1. Elaborate on the realism and literature precedent of Assumptions 3-5 to justify their inclusion.
2. Correct the PDF's non-functional links, absence of a table of contents, and alignment with TMLR formatting guidelines.
3. Improve the presentation and clarity by addressing the weaknesses comments.

**Strengths And Weaknesses:**

## Strengths
- The paper tackles a relatively unexplored problem.
- It introduces “in distribution” convergence, which appears novel for this family of methods.
- The authors provide rigorous theoretical analysis.
- Extensive numerical experiments comparing the performance to baselines are presented.

## Weaknesses
- The introduction section lacks practical motivation. Why are the results important and interesting in the specific context of streaming stochastic optimization with heavy tails and outliers?
- The paper's presentation and theoretical results are not very clear.

Next, I structure my comments and questions according to the sections.

### Section 2

In my opinion, Assumptions 3-5 need a more detailed discussion. How realistic are these assumptions? Are they introduced for the first time, or are they standard in the literature? Are there any other analogs or alternatives? Are these conditions technical and introduced to make the proof work, or do they naturally hold in some applications that motivate the work? This section seems to lack references to prior works that may have relied on some of the assumptions.

Assumption 4 seems unusual in representing the error’s distribution as a linear combination. What is \(\theta\)? Is it a parametrization of the density?

- What do the subscripts $\mathcal{I}$ and $\mathcal{O}$ for $\mathcal{D}$ denote or refer to?
- It is unclear why Assumptions 4 and 5 are separated, as both seem to impose conditions on the stochastic gradient error.
- The part on the total variation (TV) norm lacks references. Perhaps I am not an expert in this area. What is $f$ in the integral definition, and what are the necessary assumptions on it? Why was this particular measure chosen?

### Section 3

- Can the result of Theorem 1 hold for an arbitrary quantile, or does it need to be specifically chosen based on the generally unknown corruption rate $\eta$?
- The conditions on $\kappa$ and step size $\beta$ are unclear and hard to interpret. The bound on the corruption rate $\eta$ seems very restrictive concerning the condition number of the problem $\mu/L$, as the latter can be extremely small.
- How restrictive is the condition (7)? How does it compare to LMC/SGD without clipping? In general, it seems the paper could benefit from more comparisons like this.
- The discussion after Proposition 1 mentions “poor dimension dependence through $B_q$”. Why is it poor, and why is this the case? It mentions the choice $q=\log(1/\eta)$. Why do you have a choice of $q$ if it is part of the assumption? The restriction on the set $\Theta$ is quite restrictive.
- Why are the concentration properties given by Proposition 2 called “strong”?
- The formatting of Algorithm 1 could be improved. It is not very clear. I suggest adding a verbal description. What does the notation $r^{(i)}_{N/2}$ mean? Maybe I missed the definition. There is an extra break in the input description.
- The end of Section 3 mentions that
  > [28] results enjoy better dimension dependence but are less general than ours.

 How much more general is this analysis compared to prior works?

### Section 4

Why does Theorem 2 require the objective to be positive? Condition (13) looks quite obscure and hard to interpret. Can it be simplified?

### Section 5

The statement that
> In the non-strongly convex case, a constant threshold can be used since the gradient is a priori uniformly bounded

Does not seem correct, as a non-strongly convex quadratic function may not have a bounded gradient.

How does Algorithm 2 compare to adaptive clipping [1]?

The *linear regression* paragraph states that
> These thresholds provide a rough estimate of the gradient norm.

The "gradient norm" computed at which point?

### Minor Comments
- The template used is different from the official one suggested by TMLR.
- The table of contents in the PDF is absent. The links (to equations and references) starting from the second page are not working properly. Combined with the split into the main paper and supplementary material, this makes the paper very inconvenient to read and review. I recommend attaching the full paper as supplementary material on OpenReview.

___
[1] Andrew, Galen, et al. "Differentially private learning with adaptive clipping." Advances in Neural Information Processing Systems 34 (2021).

---

> ### Author Response · Authors · 2024-08-29
> **Reply to Reviewer z2UW**
>
> We thank you for your careful reading and pointing out many improvements. We had to shorten this answer for OpenReview and omitted revision details. Please see the revised paper.
>
> We added comments after the QC-SGD formula to clarify that this method is the first to grant robustness to outliers in streaming stochastic optimization in addition to heavy tails. We state the optimality of obtained bounds in our contributions and that the implementation is efficient.
> Assumption 3 corresponds to additive contamination and is compared to other works. We explain that Assumption 4 is needed for proving Markov chain convergence. We refer to other works using it and MC literature it comes from. We show Assumption 5 is in line with Assumption 1 and how it improves analogs in other works on clipped SGD.
> $\theta$ is the variable of $\mathcal{L}$. We require that the error distribution can be represented with two components, one must be diffuse with a density satisfying a minorization inequality.
>
> $I$ and $O$ are inliers and outliers. $D_I$ is the distribution of "true" samples and $D_O$ are corrupted gradients.
>
> Assumptions 4 and 5 are separated merely as an organization choice to refer to these assumptions separately on multiple occasions.
>
> $f$ in the definition of TV is such that $|f(\theta)| \leq 1$. There are no other assumptions. We use the TV because it is the most common measure of Markov chain convergence. It also enables high probability bounds as in Corollaries 1 and 2. These bounds hold for the invariant distribution and are extrapolated on $\theta_t$ thanks to Theorem 1.
>
> Theorem 1 holds for any $p \in [\eta, 1-\eta]$ such that $\kappa > 0.$ Assumption 3 requires $\eta < 1/2$. However, since $\kappa$ depends on $\mu, L$ and $\eta,$ which are unknown, we can not provide an explicit value of $p$ for which the conditions are definitely satisfied. Still, setting $p=1-\eta$ allows to show $\kappa > 0$ corresponds to $\eta^{1-1/q} \leq \mathcal{O}(\mu/(L + A_q))$ which is on par with concurrent works [28] for q=2.
>
> Condition (7) aims to be as generic as possible by not fixing p. To get an interpretation, we take $p=1-\eta$ and get the previous condition on $\eta$. This is the highest possible corruption rate in Theorem 1. This bound seems optimal because it also appears in [28]. Since this condition is mainly linked to corruption which is not in standard LMC and SGD, it is unclear how to make a comprison.
>
> The bound on $\eta$ with $\mu/L$ seems restrictive but is used in [28, Theorem E.9] for $q=2$ and seems standard. We relaxed the condition on $\beta$ in Theorem 1. If $\kappa > 0$, then we have $\kappa = \mathcal{O}(\mu)$ so with our new condition on $\beta$ and the highest possible $\eta$ with $q=2,$ we get a condition $\beta = \mathcal{O}(\mu/(\mu^2 + A_q^2) \wedge 1/L).$ $\beta \leq 1/L$ is standard in smooth optimization. The bound with $A_q$ limits the impact of noise so it doesn't cause the iteration to diverge. See the revised Theorem 1 and its comments.
>
> In general $B_q = O(\sqrt{d})$ which causes poor dimension dependence. This is due to the Euclidean norm in A5. Optimally there is no dependence on $d$ (see [28]). This is because our clipping is isotropic as opposed to filtering using the covariance matrix. The "choice" $q = \log(1/\eta)$ is for $K$-sub-Gaussian gradients. Then, Assumption 5 holds for $q\geq 1$ with $B_q \sim K\sqrt{q}$. In this case, $q$ is optimized in Proposition 1 and setting $q = \log(1/\eta)$ is optimal.
>
> Restricting to a bounded $\Theta$ is strong but we believe it is necessary for Proposition 2 when Assumption 5 hold only with finite q.
> We use the term "strong concentration" for distributions which have a finite exponential moment (i.e. sub-exponential or sub-Gaussian). This implies, for instance, the existence of finite moments of all orders.
> The notation $r^{(i)}{N/2}$ refers to the median entry of $r^{(i)}$. We added a verbal description and improved formatting of Algorithm 1.
> Our analysis is more general than [28] because we handle $q\in (1,2)$ and consider all strongly convex and also non-convex objectives. [28] focuses on mean estimation and extends to linear and logistic regression.
> Theorem 2 requires the objective is positive but was relaxed to having a lowerbound. It is needed to allow the convergence of the objective value. Condition (13) requires the set where the gradient is smaller than the estimation error to be bounded. In the alternative, the iteration may diverge as said in the comments. We added comments on condition (13) and included a simplification in big O notation.
> We corrected the statement about the non-strongly convex case.
> A. Galen et al. uses gradient clipping for differential privacy while we use it for robustness. Their quantile estimation method is ingenious. However, it has the same computational cost and would not improve the estimation.
> In linear regression, we meant the gradient norm near $\theta^\star.$
>
> https://coral-kariotta-22.tiiny.site

---

> > ### Comment · Reviewer_z2UW · 2024-09-10
> > **Additional questions**
> >
> > Dear Authors,
> >
> > Thank you for the detailed and thorough response. I recently looked through the proofs in the Appendix and had a few questions regarding some of the steps. Namely, I would like to ask for a more detailed explanation of
> >
> > 1. the second transition in upper bound for $\||\mathbb{E}[\alpha_\theta \widetilde{G}(\theta)]-\bar{\alpha}_\theta \nabla \mathcal{L}(\theta)\||$ on page 24
> >
> > 2. the upper bound used for $\mathbb{E}[\||\varepsilon_\theta\||^q]$ on page 24
> >
> > 3. the first inequality in upper bound for $\mathbb{E}\||\alpha_\theta \widetilde{G}(\theta)] - \mathbb{E}[\bar{\alpha}_\theta \nabla \mathcal{L}(\theta)]\||^2$ on page 24
> >
> > In addition, I suggest splitting the Proof of Theorem 2 into parts with the text explanation of the transitions as, in its current form, it is hard to comprehend. I believe that it is pretty important as the paper's main contributions are theoretical.
> >
> > Best regards

---

> > > ### Author Response · Authors · 2024-09-11
> > > **Reply to Reviewer z2UW**
> > >
> > > Dear reviewer,
> > >
> > > Regarding the second transition in bounding
> > >
> > > $||\mathbb{E}[\alpha_{\theta} \widetilde{G}(\theta)] - \bar{\alpha}_{\theta} \nabla \mathcal{L}(\theta)|| $
> > >
> > > , it is due to the fact that, under the event $\{||\widetilde{G}(\theta))|| >  \tau_\theta\},$ we have $0\leq \tau_{\theta}/||\widetilde{G}(\theta))|| < 1$ and therefore $|1 - \tau_{\theta}/||\widetilde{G}(\theta))|| | \leq 1$.
> > >
> > > The inequality we use on $\mathbb{E} [ ||\varepsilon_{\theta} ||^q]$ corresponds to Assumption 5. This is possible because Lemma 2 considers a fixed $\theta$.
> > >
> > > Regarding your third question, we are assuming that you meant the upper bound for $\mathbb{E} ||\alpha_{\theta} \widetilde{G}(\theta) - \mathbb{E}[\alpha_{\theta} \widetilde{G}(\theta)]||^2$. The first inequality results from the fact that we always have $||\alpha_{\theta}\widetilde{G}(\theta)|| \leq \tau_{\theta}$ since the factor $\alpha_{\theta}$ corresponds to applying the clipping threshold $\tau_{\theta}$.
> > >
> > > As a result, we have $||\alpha_{\theta} \widetilde{G}(\theta) - \mathbb{E}[\alpha_{\theta} \widetilde{G}(\theta)]||^2 \leq (2\tau_{\theta})^2 = 4\tau_{\theta}^2$.
> > >
> > > We have added these details to the manuscript to make these steps clearer. As suggested, we have also broken down the proof of Theorem 2 to simpler, more progressive and commented steps to make it easier to understand for the reader.

---

> > > > ### Comment · Reviewer_z2UW · 2024-09-12
> > > > **Acknowledgment**
> > > >
> > > > Dear Authors,
> > > >
> > > > I would like to thank you for the productive discussion, detailed answers, and incorporation of feedback.
> > > >
> > > > > we are assuming that you meant the upper bound for
> > > >
> > > > Yes, exactly; thanks for that.
> > > >
> > > > Now, the clarity and readability of the mentioned proofs seem quite better to me.
> > > >
> > > > Best regards

---

> > > > > ### Author Response · Authors · 2024-09-12
> > > > > **Thanks**
> > > > >
> > > > > Dear Reviewer,
> > > > >
> > > > > We are glad to learn that our answers have met your expectations.
> > > > > We also wish to thank you for your detailed review and constructive criticism which led to considerable improvements to our original manuscript.
> > > > >
> > > > > Best regards

---

### Review · Reviewer_kBQC · 2024-08-17

**Summary Of Contributions:**

The work proposes a new variant of clipped SGD with the choice of clipping threshold based on quantiles of stochastic gradients. The convergence of this algorithm in distribution is studied for strongly convex and non-convex problems. Numerical results complement the study.

**Audience:**

Yes

**Claims And Evidence:**

Yes

**Requested Changes:**

1. [62] and [30] do not require sub-Gaussian assumption.

2. Before [85], NY in [63] already showed how to handle heavy tailed noise by changing the mirror map.

3. "initial evidence of the robustness of clipped SGD to heavy tails was given by [87]".

4. Part 1 of Lemma 2 controls the bias of uncorrupted stochastic gradients and looks standard for for gradient clipping analysis. Perhaps adding some references would be appropriate. Same applies to Lemma 1 if it appeared in prior work.

5. Why the definition of the transition kernel is different on page 4 and on page 24. The most confusing thing for me is that on page 4 and in the discussion on 25, the transition kernel has the event A as an argument, but in the definition on page 24, it does not have it.

6. To make the paper self-contained it's better to add some formulations from [58], for instance, in the end of the proof of Theorem 1 on page 26, some condition (iii) from [58] is referenced but not stated. Definition of petit set is not given, is it even relevant for the proof?

7. How does (25) imply the result of Theorem 1? Where the measure $\pi_{\beta, p}$ is defined?

8. A better title for section 4 is perhaps "Non-convex objectives" because results of section 3 are also for smooth functions.

**Strengths And Weaknesses:**

**Strengths:**

Theoretical claims in this paper look solid. The proofs in the appendix are easy to follow except for some minor issues described below.

**Weaknesses:**

The proposed quantile clipping is an interesting algorithm. However, the motivation for its use (compared to the standard clipping) remains unclear to me. I could not find any theoretical justification, how does it compare in theory to standard clipped SGD?

Main issue: Quantiles are not computable (since the distribution is unknown) and can be hard to estimate under heavy tailed noise. Authors say in introduction that QC-SGD can be implemented using rolling quantiles, but when it comes to section 5 it turns out that Algorithm 2 is actually a different algorithm. There is no proof that it implements exactly the rule (2) that is analyzed in theory.

---

> ### Author Response · Authors · 2024-08-29
> **Reply to Reviewer kBQC**
>
> We thank you for your time spent reading and reviewing our manuscript and for your constructive comments.
>
> Our main motivation to use quantile clipping is to achieve robustness to $\eta$-corruption as defined in Assumption 3. Using quantile clipping also allows the iteration to be adaptive to the gradient variance depending on the current value of the parameter $\theta_t$. This is another advantage of quantile clipping which allows to use a looser bound on the gradient variance which depends on $\|\theta_t - \theta^\star\|$ as in Assumption 5. In constrast, previous works using standard clipping with a constant threshold required uniform bounds on the gradient variance [31, Gorbunov et al.] or could not handle the presence of corruption [81, Tsai et al.]. Finally, quantile clipping also leads to strong concentration properties for the limit distribution. We have included these additional clarifications just before the statement of our contributions.
>
> We state that our implementation uses rolling quantiles in order to estimate the quantile of the gradient norm $Q_p(\|\widetilde{G}(\theta_t, \zeta_t)\|)$. The exact value of the latter can not be obtained in general due to the data distribution being unknown. The proposed implementation aims to approximate iteration (2) while remaining computationallly efficient.
>
> We address the requested changes below:
>
> 1. We have modified the concerned sentence in our introduction to correct this mistake.
>
> 2. We have added a citation of [63] in the concerned phrase of our "Related works" section in order to give due credit for this early finding.
>
> 3. We do not understand what change is requested for the quoted sentence. Please be more explicit. Note that although its title may look unfit for this context. Reference [87] is truly the one we meant to cite here considering its content.
>
> 4. Despite searching papers in the litterature, part 1 of our Lemma 2 appears to be new in our work and we are unaware of any similar statement to it in the past. This result is enabled by the specific choice of the gradient norm quantile as clipping threshold and our mathematical analysis of this method appears to be novel. Regarding Lemma 1, there actually was a similar statement in Proposition 2 of (Dieuleveut et al. 2017). We have added these informations and further comments after the statements and proofs of Lemmas 1 and 2 in our revision.
>
> 5. The definition of the Markov transition kernel does not change but the difference between page 4 and page 24 is that, in the former, the kernel is applied to a measure whereas in the latter, it is applied to a function. We have added a step in the concerned equation on page 24 to detail how the kernel is applied to a function with the original notation. Hopefully, this clears the confusion.
>
> 6. We have added a section before the proof of Theorem 1 in order to introduce the formulations we need from [58] including the definition of a petite set. The latter is relevant for our proof because it is a hypothesis we need to verify in order to apply Theorem 15.0.1 from [58]. We have also made the verification of the requirements of condition (iii) of Theorem 15.0.1 from [58] more explicit in our proof of Theorem 1.
>
> 7. The conclusion of the proof of Theorem 1 uses the fact that the norm $\|\cdot\|V$ (defined for any positive function $V$ in the new section before the proof of Theorem 1) dominates the Total Variation norm. We have added this detail in the proof. $\pi_{\beta, p}$ is the invariant measure for $(\theta_t)$. We only establish its existence and uniqueness in Theorem 1 but explore its properties in subsequent propositions and corollaries. We have added a comment on this in the discussion following the statement of Theorem 1.
>
> 8. We have followed your recommendation and renamed section 4 as suggested.

---

### Author Response · Authors · 2024-08-29
**General answer**

We thank all reviewers and the action editor for their time reading and reviewing our manuscript. We provide detailed answers to their comments and questions in the answers below. We also revise the paper to reflect their observations and suggestions and make some improvements to original results and their presentation. We hope that our answers and the revised paper meet the standards of TMLR and remain ready to answer further questions.

Unfortunately, we had to considerably shorten our original answer to reviewer z2UW to fit OpenReview's constraint on answer length. We hope that our choice to include a link to the original full length answer we wrote will not be severely judged.

---

### Decision · Action_Editor_ExFX · 2024-09-27

**Recommendation:** Accept with minor revision

**Comment:**

The paper receives three positive reviews from experts in stochastic optimization. The reviews acknowledge the novelty of the theoretical analysis and the algorithm design, as well as the extensive experimental analysis. In particular, the paper presents interesting convergence analysis based on convergence in distributions, which has potential to be applied to other convergence analysis. The theoretical analysis seems to be rigorous. Efficient implementation based on rolling quantiles is also given with experimental verification.

Suggestion: it may be better to remove the contents part in the paper.

**Audience:**

Machine learning often involves heavy-tailed data. The paper proposes a new approach to handle heavy-tailed data, which is interesting to the TMLR community.

**Claims And Evidence:**

The paper proposes a new stochastic gradient algorithm based on the clipping operator, where the clipping factor is chosen according to the quantile of  stochastic gradients. The motivation is to improve the robustness of the algorithm to handle heavy-tailed samples. The paper provides theoretical analysis for convex and nonconvex problems. The paper also gives an efficient implementation of the algorithm based on rolling quantiles, and verifies the efficiency by extensive experimental comparisons.